

# The role of cyclones and PV cutoffs for the occurrence of unusually long wet spells in Europe

Matthias Röthlisberger[1], Barbara Scherrer[1], Andries Jan de Vries[1,a], and Raphael Portmann[1]

[1]Institute for Atmospheric and Climate Science, ETH Zürich, Zürich, Switzerland
[a]now at Institute of Earth Surface Dynamics, University of Lausanne, Lausanne, Switzerland

*Correspondence to*: Matthias Röthlisberger (matthias.roethlisberger@env.ethz.ch)

**Abstract.** The synoptic dynamics leading to the longest wet spells in Europe are so far poorly investigated, despite these events' potentially large societal impacts. Here we examine the role of cyclones and PV cutoffs for unusually long wet spells in Europe, defined as the 20 longest uninterrupted periods with at least 5 mm daily accumulated precipitation at each ERA-Interim grid point in Europe (this set of spells is hereafter referred to as $S_{20}$). The $S_{20}$ occur predominantly in summer over the eastern continent, in winter over the North Atlantic, in winter or fall over the Atlantic coast, and in fall over the Mediterranean and European inland seas. Four case studies reveal distinct archetypal synoptic storylines for long wet spells: (a) A seven-day wet spell near Moscow, Russia, is associated with a single slow-moving cutoff-cyclone couple; (b) a 15-day wet spell in Norway features a total of nine rapidly passing extratropical cyclones and illustrates serial cyclone clustering as a second storyline; (c) a 12-day wet spell in Tuscany, Italy, is associated with a single but very large cutoff-complex, which is replenished multiple times by a sequence of recurrent anticyclonic wave breaking events over the North Atlantic and western Europe; and (d) a 17-day wet spell in the Balkans features intermittent periods of diurnal convection in an environment of weak synoptic forcing and recurrent passages of cutoffs and thus also highlights the role of diurnal convection for long wet spells over land. A systematic analysis of cyclone and cutoff occurrences during the $S_{20}$ reveals considerable spatial variability in their respective role for the $S_{20}$. For instance, cyclones and cutoffs are present anywhere between 10% and 90%, and 20% and 70% of the $S_{20}$ time steps, respectively, depending on the geographical region. However, overall both cyclones and cutoffs appear in larger number and at a higher rate during the $S_{20}$ compared to climatology. Furthermore, in the Mediterranean, the cutoffs and cyclones are significantly more persistent during the $S_{20}$ compared to climatology. Our study thus documents for the first time the palette of synoptic storylines accompanying unusually long wet spells across Europe, which is a prerequisite for developing an understanding of how these events might change in a warming climate and for evaluating the ability of climate models to realistically simulate the synoptic processes relevant to these events.

## 1 Introduction

The nature of precipitation is episodic. In Europe, precipitation episodes vary strongly regarding peak and mean precipitation rates as well as duration. Unusually large peak precipitation rates, often termed *precipitation extremes*, have received much



attention in the atmospheric dynamics community, and the synoptic-scale dynamical processes leading to precipitation extremes have been studied in detail for precipitation extremes on a wide range of time-scales, from minutes (e.g. Lenderink and Van Meijgaard, 2010) to seasons (e.g., Flaounas et al., 2020). Research on extreme precipitation is often motivated by these events' relevance for a wide range of natural hazards such as landslides and flooding (e.g., Moore et al., 2012; Rössler

et al., 2014; de Vries, 2021). However, studies on individual high-impact events often report an extended duration of the event, for instance four consecutive days of heavy rain leading to the record floods in Germany in early summer 2013 (Grams et al., 2014), a sequence of multi-day precipitation episodes causing the devastating Pakistan flood in summer 2010 (Martius et al., 2013), or a temporal clustering of wet periods leading to an emergency spill-over of California's Oroville Dam in February 2017 and mass evacuation of the population living downstream (White et al., 2019; Moore et al., 2020). Clearly, in addition

to peak intensities, the temporal characteristics of precipitation episodes are thus highly relevant too.

Previous studies have assessed these characteristics often by focusing on continuous or quasi-continuous episodes of precipitation, so called *wet spells* (e.g., Berger and Goossens, 1983; Schmidli and Frei, 2005; Tolika and Maheras, 2005; Zolina et al., 2010, 2013). For example, Zolina et al. (2013) found that the mean duration of wet spells in Europe over the last 60

years is between 1.5 days over the Ukraine and southern Russia and 5 days over the Scandinavian Atlantic coast. They reported that the longest wet spells last between 3 (eastern Europe) and 12 days (Scandinavia). It should be noted that these durations depend on the specific definition of wet spells. Furthermore, they found an increasing duration of wet spells during the last 60 years, especially over northern and central Europe, and a slightly decreasing duration over southern Europe during the cold seasons. In Scandinavia and eastern Europe, the duration of wet spells decreased significantly during the warmer seasons.

Schmidli and Frei, (2005) focused on observational data from Switzerland and identified seasonally and regionally varying trends in the duration of the longest wet spells per year. However, despite a considerable body of literature on statistical characteristics of wet spells, the synoptic-scale dynamical mechanisms leading to precipitation episodes with an unusually long duration – typically one to two weeks – have so far not been investigated in detail.

For midlatitude extreme precipitation events on timescales from a few hours to a few days it is well known that different large-scale weather systems can serve as dynamical precursors. These are extratropical cyclones (Ulbrich et al., 2003; Field and Wood, 2007; Pfahl and Wernli, 2012), fronts (Catto and Pfahl, 2013; Rüdisühli et al., 2020), so-called "warm conveyor belts" (WCBs; Pfahl et al., 2014), long-range horizontal moisture transport (Winschall et al., 2014), often in the form of so-called "atmospheric rivers" (Zhu and Newell, 1998; Ralph et al., 2004; Lavers and Villarini, 2013), upper-level short-wave troughs

and cutoffs (Massacand et al., 1998; Martius et al., 2006), and even atmospheric blocks (Sousa et al., 2017; Lenggenhager and Martius, 2019). These weather systems usually do not occur in isolation, but given their dynamical relationship and interactions, it is rather their combination that leads to an extreme precipitation event (de Vries, 2021). For instance, a narrow upper-level trough is often linked to an elongated surface cold front and intense poleward moisture transport, along which





frontal-wave cyclogenesis and the subsequent formation of a WCB often occurs, and all of these weather systems act in concert
to produce the heavy precipitation.

For unusually long-lasting wet spells, it is much less clear how and in association with which weather systems they form. Only
few studies investigated the causes of long-lasting precipitation episodes and mostly focused on multi-day heavy precipitation
events rather than (potentially even much longer) wet spells. Moore et al. (2021) investigated multi-day (i.e., longer than 3
day) episodes of heavy precipitation along the North American west coast and argued that, on a general level, multi-day heavy
precipitation events either occur when individual rain producing weather systems stall or when multiple such weather systems
occur in a serially clustered manner.

In particular in the North Atlantic region, serial clustering of extratropical cyclones is well documented (Mailier et al., 2006;
Pinto et al., 2014; Priestley et al., 2017a, b; Dacre and Pinto, 2020). This phenomenon occurs predominantly at the downstream
end of the North Atlantic storm track (Mailier et al., 2006; Dacre and Pinto, 2020) and arises over western Europe preferentially
when the North Atlantic jet is extended towards Europe, remains at a similar latitude for a prolonged period (typically more
than a week) and thereby steers entire cyclone families (i.e., primary cyclones as well as frontal wave cyclones forming on
trailing cold fronts of the primary cyclones) into the same region (Pinto et al., 2014; Priestley et al., 2017a; Dacre and Pinto,
2020). During such clustering periods, the jet is kept in place by momentum fluxes arising from cyclonic and anticyclonic
Rossby wave breaking on the poleward and equatorward side of the jet, respectively. The process of serial cyclone clustering
is clearly relevant for long precipitation episodes, as both the extremely wet winter 2013/14 in the United Kingdom and a large
set of multi-day heavy precipitation in California have been related to serial cyclone clustering (Priestley et al., 2017b; Moore
et al., 2021).


The stalling of individual cyclones as well as upper-level flow features as the cause of long-lasting heavy precipitation events
has also been documented in multiple cases, e.g., for an event in Spain (Doswell et al., 1998) when a single slow-moving
cyclone associated with an upper-level cutoff caused a seven-day heavy precipitation event. Cutoffs often form from narrow
filaments of stratospheric potential vorticity (PV), so-called PV streamers (Appenzeller and Davies, 1992). Upper-level PV
streamers and cutoffs are accompanied by a cyclonic wind field with a far field effect down to the lower troposphere (Hoskins
et al., 1985). Below such PV features the static stability is reduced, which can contribute to the occurrence of deep convection
and heavy precipitation (Massacand et al., 1998; Romero et al., 1999; Martius et al., 2006; Portmann et al., 2018; Moore et al.,
2019). Moreover, cutoffs in a baroclinic zone are associated with quasi-geostrophic forcing for ascent on their downstream
side, cloud formation and precipitation, even if they remain stationary. Since some cutoffs are relatively long-lived and
stationary, they can play an essential role in the formation of multi-day precipitation extremes (e.g. Grams et al., 2014). Thus,
individual unusually stationary PV streamers or cutoffs also need to be considered as potential dynamical precursors of
unusually long-lasting wet spells.



Furthermore, some recent studies have highlighted the role of recurrent upper-level dynamics for long-lasting wet periods. For
example Lenggenhager et al. (2019) documented a case where recurrent PV streamer formation in association with an
atmospheric block induced a prolonged wet period and flooding on the Alpine South side. Along a similar line of arguments
Ali et al. (2021) showed that recurrent synoptic-scale Rossby wave packets [i.e., a succession of wave packets that are each in
phase such that multiple troughs form repeatedly in the same region, see Röthlisberger et al. (2019) for details] significantly
increase the duration of summer wet spells in parts of Central Europe and Iberia.


In summary, previous research focused on the statistics of wet spells or on the dynamics of short-term to multi-day heavy
precipitation events, but rarely on the dynamical mechanisms responsible for unusually long-lasting wet spells. However,
considering the potentially high societal impact of unusually long wet spells and the reported trend of increasing wet spell
duration, it is crucial to improve our understanding of the responsible dynamical processes that lead to these events. Previous
studies clearly identified serial clustering of extratropical cyclones and stalling individual or recurrent upper-level cyclonic
flow features (e.g., PV cutoffs) as key for multi-day heavy precipitation events. Here we hypothesize that extratropical cyclones
and PV cutoffs also play an important role for the quasi-continuous precipitation during unusually long wet spells. Therefore,
we pragmatically choose these two weather systems and examine their role for the formation of unusually long wet spells in
Europe. Specifically, the purpose of this study is to identify unusually long wet spells in Europe using ERA-Interim reanalysis
data (Dee et al., 2011), and to quantify the occurrence of cyclones and cutoffs during these spells. Hereby, we focus on the 20
longest wet spells (as per our definition of wet spells, see below) at each ERA-Interim grid point in Europe and address the
following research questions:

1) How do the duration, accumulated precipitation, average precipitation rate and seasonality of the longest wet spells
   vary across Europe?

2) What synoptic storylines accompany these unusually long wet spells and how are cyclones and cutoffs involved in
   the generation of these wet spells?

3) How do the roles of cyclones and cutoffs in these synoptic storylines vary across Europe?

4) Where and how do the characteristics of cyclones and cutoffs during the longest wet spells differ significantly from
   climatology?


The structure of the paper is as follows. In Section 2 we introduce the data used in this study and elaborate on our definition
of wet spells as well as on the identification algorithms for cyclones and PV cutoffs. The results of this study are presented in
Section 3, where we first discuss climatological characteristics of the longest wet spells in Europe (Section 3.1) and then
present four case studies of unusually long wet spells at different locations, which each feature a distinct archetypal synoptic
storylines (Section 3.2) and finally address research questions 3) and 4), by systematically analysing the occurrence of cyclones





and PV cutoffs during the 20 longest spells at each grid point (Section 3.3). The paper ends with a discussion of these results (Section 4) as well as a summary and the conclusions of this study (Section 5).

## 2 Data and Methods

### 2.1 ERA-Interim

We use data from the European Centre for Medium-Range Weather Forecasts (ECMWF) ERA-Interim re-analysis from 1 January 1979 to 31 December 2018 for identifying wet spells and for the additional weather system investigations. ERA-Interim data has originally been produced with a T255 resolution and is interpolated here to a regular 1° latitude by 1° longitude grid. In ERA-Interim, precipitation is not an assimilated variable but rather stems from short-range model forecasts, with lead times of 6-18 hours, and is thus subject to model limitations and forecast errors. Pfahl and Wernli (2012) showed that indeed

the intensity of sub-daily ERA-Interim precipitation extremes is often underestimated compared to satellite observation-based estimates, in particular in the tropics. However, the timing and location of intense precipitation is well represented in ERA-Interim in comparison with satellite observations (Pfahl and Wernli, 2012).

### 2.2 Definition of the longest wet spells

Here we focus on episodes with uninterrupted and significant, but not necessarily extreme precipitation. We therefore define a wet spell $S$ as a sequence of consecutive days, each with at least 5 mm accumulated precipitation (large-scale plus convective). At each grid point with coordinates $(x, y)$ the 20 longest wet spells are considered, and this set of spells is referred to as $\mathbf{S}_{20,}(x, y)$, where the $n$-th longest spell (i.e., an individual event, with a duration, starting date, accumulated precipitation value etc.) is denoted as $S_n(x, y)$. The coordinate specification $(x, y)$ is omitted wherever possible without loss of clarity. Note

that alternative definitions of wet spells, for example based on a different daily precipitation threshold or by allowing short gaps between precipitation episodes, would yield distinct sets of events for the top 20 longest wet spells per grid points. Thus, the identification of "the longest wet spells" is to some degree subjective, but the $\mathbf{S}_{20}$ as defined here certainly classify as unusually long wet spells, which, due to the relatively high threshold for daily precipitation, also have the potential to lead to societal impacts. Furthermore, we tested the sensitivity of our results to different sample sizes (top 5 and 10 longest wet spells)

and found no qualitative differences, but for these smaller samples the results were statistically less robust.

### 2.3 Identification of cyclones and PV cutoffs

To identify cyclones the identification scheme of Wernli and Schwierz (2006) is employed, which first identifies local sea level pressure (SLP) minima and then finds for each local SLP minimum the outermost SLP contour that encloses only the respective minimum. This procedure yields a binary cyclone field with individual cyclone objects. The algorithm furthermore



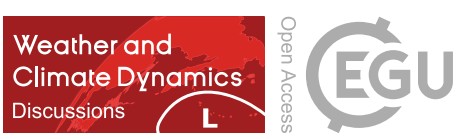

tracks these individual objects in time based on the location of the cyclone center (i.e., SLP minimum) [see Sprenger et al. (2017) for details of the tracking] and only cyclones with a lifetime of at least 24 hours are considered in this study.

PV cutoffs are identified and tracked according to Portmann et al. (2021). This method first identifies PV cutoffs on individual isentropic surfaces between 275-360 K with 5 K intervals as isolated regions with PV values above 2 PVU following Wernli

and Sprenger (2007). Then, PV cutoffs are defined as 3-dimensional objects that are vertically connected to the stratosphere. This largely removes high PV features of tropospheric origin that arise due to e.g. surface friction or diabatic heating. PV cutoffs are tracked in time based on Lagrangian air parcel trajectories and only cutoffs with a lifetime of at least 24 hours are retained [for details of the identification and tracking see Portmann et al. (2021)]. A key advantage of this approach compared to earlier PV cutoff identification and tracking routines is that it is independent of the selection of single vertical levels. For

subsequent analyses we use the projection of the 3-dimensional cutoff objects onto the Earth's surface, which, as for cyclones, yields binary cutoff fields with individual cutoff objects.

### 2.4 Quantifying the characteristics of cyclones and PV cutoffs during the $\mathcal{S}_{20}$

To quantify the characteristics of cyclones/cutoffs occurring during the $\mathcal{S}_{20}(x, y)$ we consider for each grid point $(x, y)$

cyclones and cutoffs whose masks overlapped with a circle of 400 km radius around $(x, y)$ during at least one 6-hourly time step of the $\mathcal{S}_{20}(x, y)$. For such cyclones and cutoffs we use the terminology "occurring at $(x, y)$" hereafter for simplicity. We compute four quantities for these cyclones and cutoffs:

1)    The "cyclone/cutoff fraction", which corresponds to the fraction of all $\mathcal{S}_{20}$ time steps with cyclones/cutoffs occurring

180          at $(x, y)$. This quantity is hereafter referred to as $F_f(x, y)$, whereby $f$ corresponds to the respective feature (cyclones or cutoffs).

2)    The "number of distinct cyclones/cutoffs" per spell, $N_f(x, y)$.

3)    The "cyclone/cutoff period", $P_f(x, y)$, which is the average duration between the arrival of distinct cyclones/cutoffs within the 400 km radius around $(x, y)$, computed as the summed duration of the $\mathcal{S}_{20}(x, y)$ divided by $N_f(x, y)$.

4)    The "cyclone/cutoff residence time", $R_f(x, y)$, which is the average duration during which distinct cyclones/cutoffs overlap with the 400 km radius around $(x, y)$.

Note that in the computation of these metrices we consider cyclones and cutoffs also if they overlap with the respective circle only during a short period, e.g., a single time step. Thus, in particular the residence time should be interpreted with such

situations in mind. Furthermore, metrices $P$ and $R$ allow for distinguishing between recurrent features and stalling features in





the following way: Recurrent cyclones or cutoffs manifest themselves with a short cyclone/cutoff period ($P$), while long cyclone/cutoff residence times ($R$) are expected for stalling cyclones and cutoffs.

In Section 3.3 we compare the four metrics to respective "climatological" values, which are constructed in the following way:
At each grid point $(x, y)$ we identify the days of the year during which at least one of the $\boldsymbol{S}_{20}(x, y)$ occurred. Then, we identify all cyclones/cutoffs that occurred at $(x, y)$ during these days of the year in any year from 1979 to 2018. From this set of cyclones/cutoffs we then compute "climatological values" for the four metrics, in exactly the same fashion as described above. This procedure implies in particular that the days of the year which are considered for constructing the climatological values differs for each grid point, which is necessary because the seasonality of the $\boldsymbol{S}_{20}$ (i.e., the days of the year on which the $\boldsymbol{S}_{20}$
occur) also differs from grid point to grid point. Based on these climatological values we compute anomalies for all four metrices.

The statistical significance of these anomalies is assessed based on a Monte Carlo approach, in close analogy to that of Röthlisberger et al. (2016). The procedure is detailed here exemplarily for the cyclone fraction, $F_{cyclone}(x, y)$, but is applied
in exactly analogous fashion for all four metrics and both weather features. First, the occurrence of the spells is randomized by shuffling the years of the $\boldsymbol{S}_{20}(x, y)$, while retaining the actual months and days of these spells. Then, using these randomized spell dates, a randomized cyclone fraction is computed exactly as the true cyclone fraction $F_{cyclone}(x, y)$. This process is repeated 1000 times and two-sided p-values are estimated through comparing the true cyclone fraction with the distribution of randomized cyclone fractions. A p-value of zero is assigned if the observed cyclone fraction lies outside the range of the 1000
randomized cyclone fractions. Note that the randomization through the shuffling of the years of the $\boldsymbol{S}_{20}(x, y)$ conveniently circumnavigates issues arising from the varying seasonality of the $\boldsymbol{S}_{20}(x, y)$. In this study, anomalies are deemed significant at a grid point wise significance level of 0.01.

## 3    Results

### 3.1 Climatological characteristics of the longest wet spells

We begin by discussing basic characteristics of the $\boldsymbol{S}_{20}$ and their geographical variations. In many regions of Europe, the average duration of the $\boldsymbol{S}_{20}$ is on the order of four to seven days (Fig. 1a). The $\boldsymbol{S}_{20}$ are longest predominantly along the Atlantic coastal regions as well as in areas of elevated topography, and shortest over the Barents, Black and Baltic Seas. Mean durations range from less than 4 days over parts of northern Scandinavia, Crimea, Germany, and Poland to more than two weeks in western Norway (Fig 1a). The overall longest European ERA-Interim wet spell occurred at 6°E/62°N, about 300 km north of
Bergen, Norway (not shown). It started on 9 September 2018, lasted for 28 days, and finally ended on 6 October 2018.

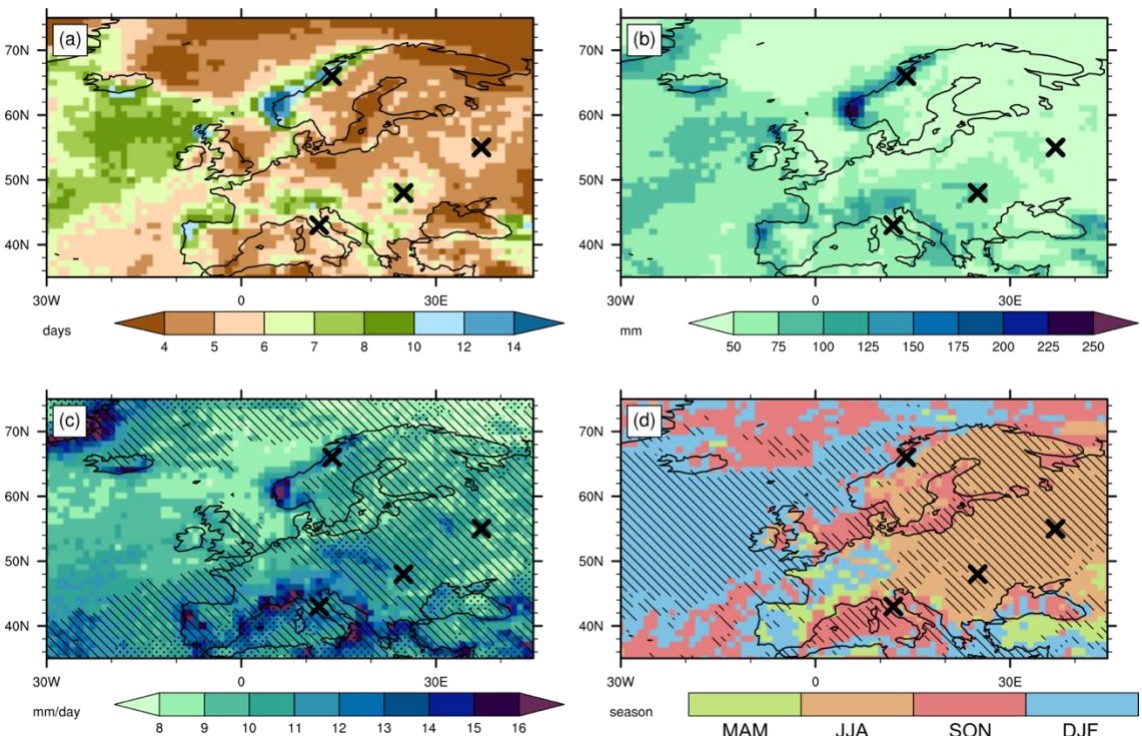

**Figure 1.** Average characteristics of the 20 longest wet spells at every ERA-Interim grid point ($S_{20}$). (a) average duration, (b) average accumulated precipitation, and (c) average precipitation rate; (d) the season in which the largest fraction of the $S_{20}$ start. Hatching and stippling in (c) depict regions where the average precipitation rate during the $S_{20}$ exceeds the 95[th] and 97.5[th] percentile, respectively, of all ERA-Interim wet days, defined as days with > 0.1 mm precipitation. Hatching in (d) shows regions where at least 12 of the $S_{20}$ start in the same season. Crosses identify grid points for which the $S_{20}$ are further examined in Figs. 2–7.

The accumulated precipitation during the $S_{20}$ co-varies with their average duration (Fig. 1a,b), and largest average accumulations are found in Norway, north-western Iberia and Scotland (>200 mm per spell). The largest precipitation accumulation during a single wet spell again occurred in western Norway, at 6°E/61°N in January 1989 (not shown). The spell lasted for 24 days, and the accumulated ERA-Interim precipitation amounted to 471 mm. The average daily precipitation rate during the $S_{20}$ exceeds 5 mm day$^{-1}$ by definition (Fig. 1c), and its spatial variability differs somewhat from those of the mean $S_{20}$ duration and accumulated precipitation. Largest daily precipitation rates during the $S_{20}$ are again found along the Iberian Atlantic coast and western Norway, with values in excess of 15 mm day$^{-1}$, but also in Mediterranean coastal regions. Average daily precipitation rates during the $S_{20}$ are locally anomalous, although not overly extreme. In most regions of Europe, they exceed the 95[th] percentile of the accumulated daily precipitation during all wet days (defined here as days with > 0.1 mm accumulated precipitation) but are below the 97.5[th] percentile (Fig. 1c), implying that between 1 out of 20 and 1 out of 40 wet days feature comparable precipitation accumulations as the ones observed on average during the $S_{20}$. This result underlines that the $S_{20}$ constitute a set of potentially high-impact precipitation episodes that is different from heavy precipitation events commonly identified based on very high (≥99[th]) percentiles of (sub)daily precipitation in previous studies (Pfahl and Wernli, 2012; Lenggenhager and Martius, 2019; Moore et al., 2021; de Vries, 2021).

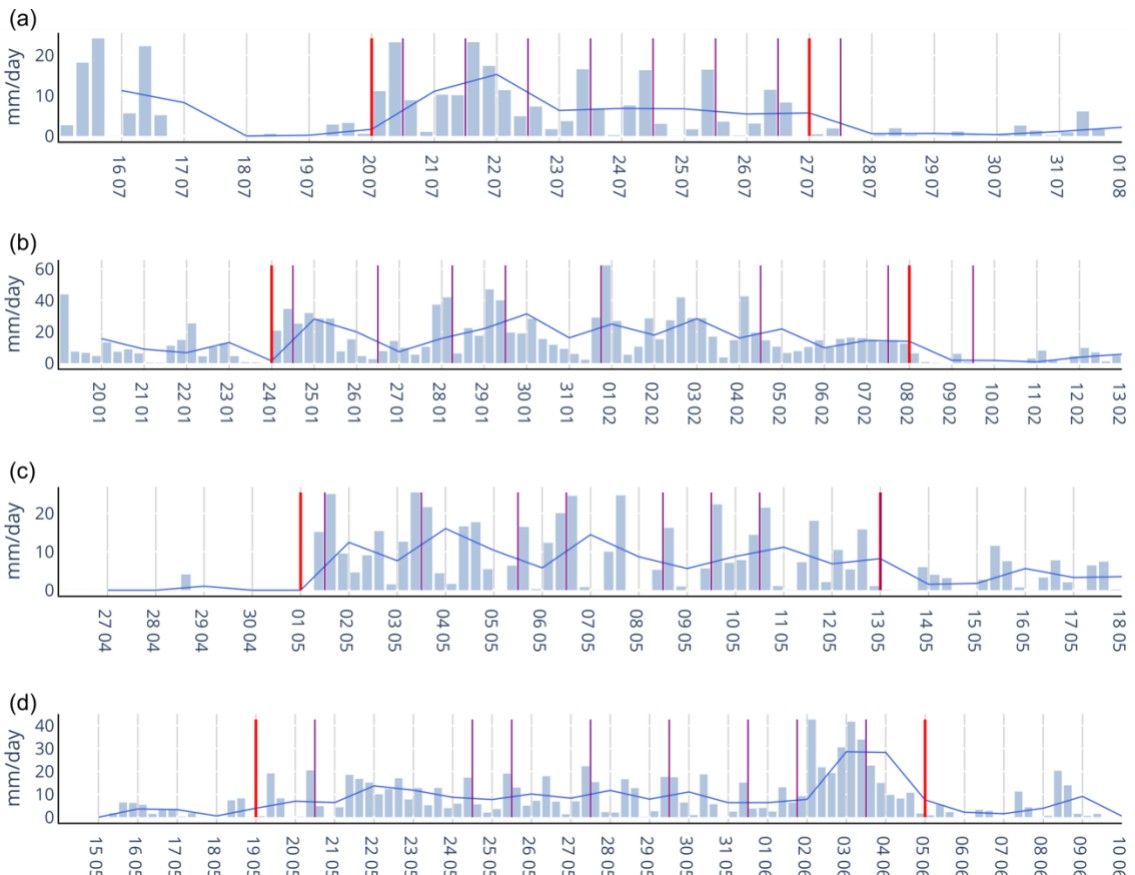

**Figure 2.** Time evolution of the wet spells illustrated in Figs. 3–6. The panels show the precipitation evolution during the longest wet spell at (a) 37°E/57°N, near Moscow, Russia in 2013, (b) at 14°E/66°N, in Norway in 1989, (c) at 12°E/43°N in Tuscany, Italy in 2018 and (d) at 25°E/48°N, in Romania in 1988. Bars depict 6-hourly precipitation (multiplied by four) while the bold blue line depicts daily precipitation accumulations. Daily precipitation is depicted for each day at 00:00 UTC and refers to the precipitation accumulation over the preceding 24 hours. Red bars mark the onset and termination of each spell, while purple lines indicate the times for which fields are shown in Figs. 3–6. Light vertical lines locate 00:00 UTC of each day.

The seasonality of the $S_{20}$ exhibits a clear spatial pattern. Over the North Atlantic and the eastern Mediterranean, most of the $S_{20}$ occur in winter (December–February, DJF), and likewise in southern Iberia and southwestern France. Over several coastal seas (the north-western Mediterranean, the North Sea, the Gulf of Bothnia and the Atlantic coastal seas near Iberia), the $S_{20}$ occur predominantly in fall (September–November, SON), while over continental eastern Europe (including Sweden, Finland, and western Russia), the vast majority of the $S_{20}$ happen during summer (June–August, JJA). At most grid points in central Europe, though, no season dominates the occurrence of the $S_{20}$. In a spatially aggregated sense, spring (March–May, MAM) has by far the lowest number of spells as only 12.8% of the $S_{20}$ spells over land and ocean occurred in MAM, followed by 22.2% in JJA, 29.1% in SON and 35.9% in DJF. When only considering land grid points these contributions change markedly, with 17.6% of land $S_{20}$ spells occurring in MAM, 46.4% in JJA, 19.0% in SON and 17.0% in DJF. Next, we illustrate the





palette of synoptic storylines of unusually long wet spells with four case studies of selected wet spells in different regions and seasons.

## 3.2 Case studies of selected unusually long wet spells

### 3.2.1 Long wet spell in western Russia – Quasi-stationary cutoff-cyclone couple

At 37°E/55°N, close to Moscow, Russia, the longest wet spell [$S_1(37°E, 55°N)$] occurred in July 2013 and lasted for 7 days (Fig. 2a). An examination of the large-scale flow pattern reveals a very persistent upper-level PV cutoff with a surface cyclone underneath during the entire spell (Fig. 3). At 12:00 UTC on 20 July, the first day of the spell, the cutoff C1 was located west

of Moscow, reaching from 75°N southward to the Black Sea, with a weak surface cyclone L1 to its east (Fig. 3a). Precipitation occurred at the southeastern flank of the cutoff. Closer analysis shows that precipitation was primarily convective at 37°E/55°N (visible, e.g., from the diurnal pattern in precipitation at 37°E/55°N, Fig. 2a), most likely supported by quasi-geostrophic forcing for ascent and the decreased tropospheric static stability induced by the upper-level cutoff. Within the next four days, little changed in this synoptic configuration over western Russia, although from 21 July onwards, a second cutoff-cyclone pair

(L2, C2) formed over the British Isles, resulting in a large-scale omega-type blocking pattern (Fig. 3b–e). The surface cyclone L1 weakened considerably between 23 and 25 July, when its local sea level pressure minimum was no longer identified as a cyclone by the cyclone identification scheme (Fig. 3f). However, it re-appeared as an identified cyclone at 12:00 UTC on 26 July (labelled L1* in Fig. 3g,h) and propagated eastward between 26 and 27 July, in tandem with the persisting cutoff C1 (Fig. 3f–h), which ultimately terminated the wet spell at 37°E/55°N on 27 July (Fig. 3h). Throughout the entire period, precipitation

during daytime (06:00-18:00 UTC) strongly exceeded precipitation during nighttime (Fig. 2a), suggesting primarily convective precipitation throughout the entire period. This case illustrates a first synoptic storyline of unusually long-lasting wet spells, in which a single quasi-stationary upper-level cutoff–surface cyclone couple repeatedly produces substantial precipitation in the same regions, by providing quasi-geostrophic forcing for ascent, destabilization of the troposphere underneath and thereby fosters diurnal convection (Hoskins et al., 1985; Portmann et al., 2018).


Extending the synoptic analysis to the 20 longest wet spells at the same grid point [$\boldsymbol{S}_{20}(37°E, 55°N)$] reveals considerable variability in the cyclone and cutoff characteristics across these events, but also some similarities. The PV composite on 310 K for all $\boldsymbol{S}_{20}(37°E, 55°N)$ time steps shows a high PV anomaly west of 37°E/55°N (Fig. 7a), reminiscent of a cyclonic wave breaking event (Thorncroft et al., 1993) and with similarities to the upper-level flow pattern during the spell $S_1(37°E, 55°N)$

(Fig. 3). During 17 of the $\boldsymbol{S}_{20}(37°E, 55°N)$ at least one cutoff occurred (one spell featured even four distinct cutoffs), with an $N_{cutoff}(37°E, 55°N)$ of 1.65, and $F_{cutoff}(37°E, 55°N)$ equal to 0.38. Similarly, for cyclones we also find between zero and


**Figure 3.** Synoptic evolution of the longest wet spell near Moscow, Russia [$S_1(37°E, 55°N)$]. Shading depicts total precipitation accumulated over the preceding 6 hours. Light red contours show the 2 PVU dynamical tropopause on 320 K, while black contours show SLP (contour interval of 4 hPa). Cyclones are depicted by blue stippling and the red hatching depicts the projection of cutoff objects (see methods for details), whose outline therefore does not need to be co-located with the 2 PVU contour on 320 K. The black cross in all panels marks the grid point at 37°E/55°N. Labels help to identify the features discussed in the main text. Panels are shown daily at 12:00 UTC from 20 to 27 July 2013.

four distinct cyclones per spell, and $N_{cyclone}(37°E, 55°N)$ equals 1.9 and $F_{cyclone}(37°E, 55°N)$ is 0.64. Furthermore, the residence times of cyclones and cutoffs during the $S_{20}$ ($R_{cyclone}$ and $R_{cutoff}$) at 37°E/55°N are 1.59 and 1.1 days, respectively.



### 3.2.2 Long wet spell in central Norway – sequence of cyclones

A contrasting synoptic storyline is evident for the longest wet spell at 14°E/66°N [$S_1(14°E, 66°N)$], in Norway (Fig. 4). This wet spell occurred between 24 January and 7 February 1989 and thus featured 15 continuous days with more than 5 mm
accumulated ERA-Interim precipitation (Fig. 2b). At 12:00 UTC on 24 January 1989 (Fig. 4a), the large-scale flow over the North Atlantic was dominated by a broad trough, with a large surface cyclone L1 with two centers (local SLP minima) north of Iceland and east of the southern tip of Greenland. Over Europe, an amplified ridge R1 was present, but at its northern fringe a low-level shortwave trough (which can be regarded as the warm frontal extension of the cyclone north of Iceland) induced onshore flow and precipitation around 14°E/66°N (Fig. 4a) and started the wet spell there. Within the next two days (25–26
January), cyclone L1 moved eastwards, and three new SLP minima developed over the North Atlantic (contained in L2 and L3, Fig. 4b). The grid point at 14°E, 66°N was located continuously in westerly flow, and precipitation fell in association with two shortwave troughs propagating across Scandinavia in rapid succession (the latter of the two is visible in Fig. 4b as an upper-level PV filament between the British Isles and Scandinavia). Within the next five days (Fig. 4b–e) cyclones L3–L5 formed over the western North Atlantic, rapidly propagated into the Norwegian Sea, and produced significant precipitation at
14°E/66°N. From 4 February onwards, the upper-level flow was remarkably zonal for three days, leading to continuous onshore flow towards 14°E/66°N, at the southern fringe of cyclones L6–L8, which passed to the north of the considered grid point (Fig. 4f,g). Finally, on 7 February, cyclone L9 developed south of Greenland and followed a more meridional track (Fig. 4g,h). L9 rapidly deepened until 9 February in association with upper-level cyclonic wave breaking (Fig. 4h). Rapid upper-level ridge-formation (R3) occurred downstream, presumably aided by diabatic processes occurring in L9's strong warm conveyor
belt (not shown). The formation of R3 interrupted the predominantly zonal flow over the eastern North Atlantic and thereby terminated the wet spell. The few PV cutoffs were mostly small and filamentous and their influence on the wet spell thus seems less obvious.

In summary, the example of $S_1(14°E, 66°N)$ illustrates a second archetypal synoptic storyline for unusually long wet spells,
in which a sequence of cyclones cross the same region in rapid succession. Hereby the moist North Atlantic airmasses impinging on the western Norwegian mountains conceivably generated orographic precipitation and thereby ensured the uninterrupted formation of precipitation, in particular during transition periods between individual cyclones. Moreover, the synoptic configuration of this spell is thus reminiscent of the North Pacific "zonal jet configuration" of Moore et al., (2021), within which numerous long-lasting heavy precipitation events in Northern California occurred. These authors emphasized the
pivotal role of landfalling atmospheric rivers for long-lasting heavy precipitation events occurring in such a flow configuration. In this study we focus on cyclones and PV cutoffs and thus leave exploring the role of atmospheric rivers for the longest European wet spells to future work. Furthermore, it is noteworthy that a total of nine distinct cyclones identified by the Wernli and Schwierz (2006) algorithm were involved in the initiation, continuation and termination of $S_1(14°E, 66°N)$, which is in stark contrast to the synoptic evolution of the previously discussed case near Moscow [$S_1(37°E, 55°N)$], in which only two
objectively identified cyclones and a single cutoff appeared to be relevant.

**Figure 4.** As Fig. 3 but for the longest wet spell at 14°E/66°N in Norway [$S_1(14°E, 66°N)$]. The light red lines depict 2 PVU on 310 K. The valid time of each panel is shown in the top left.

We next extend the analysis of cyclone occurrences to the entire set of $S_{20}(14°E, 66°N)$ (Fig. 7b). During these events, the composite large-scale upper-level flow was predominantly zonal over the North Atlantic (Fig. 7b), steering cyclones in a zonal direction across the northern North Atlantic, such that most of them cross the 14°E meridian somewhat north of 66°N. Thirteen of the $S_{20}(14°E, 66°N)$ featured five or more distinct cyclones, while the remaining spells featured between two and four



distinct cyclones. For the $S_{20}(14°E, 66°N)$ we find an $N_{cyclone}(14°E, 66°N)$ of 5.15, a $F_{cyclone}(14°E, 66°N)$ value of 0.51

and $R_{cyclone}(14°E, 66°N)$ equals 1.04 days, which underline the contrasting characteristics of cyclones affecting the $S_{20}$ at

14°E/66°N (numerous, recurrent, fast moving) and at 37°E/55°N (few and stationary), despite comparable cyclone fractions

at these two grid points (0.51 vs. 0.64). Based on synoptic analyses of several of the $S_{20}(14°E, 66°N)$, PV cutoffs seemed to

be less relevant to the $S_{20}(14°E, 66°N)$ than cyclones and are not discussed here (see also Section 3.3).

### 3.2.3 Long wet spell in Tuscany – Recurrent wave breaking and cutoff replenishment

A third archetypal synoptic storyline occurred during the longest wet spell at 12°E/43°N, in Tuscany, Italy [$S_1(12°E, 43°N)$;

Figs. 2c and 5]. This 12-day wet spell occurred in association with a large cutoff-complex over the Mediterranean that first

formed after an anticyclonic wave breaking over the North Atlantic/Europe (S1 in Fig. 5a) and was then replenished multiple

times by a sequence of further wave breaking events occurring in a similar location (Fig. 5a-e). At 12:00 UTC on 1 May 2018

(Fig. 5a), the PV-streamer S1 was located over western Europe and substantial precipitation fell to its east where quasi-

geostrophic forcing for ascent is expected to be large. At its southern fringe, cyclone L1 started tracking north-eastward. At

the same time, the incipient streamer S2 was already apparent west of the UK (Fig 5a). Within the next two days streamer S1

formed cutoff C1, while S2 developed into a next elongated PV-filament that, on 320 K, reached all the way to Morocco (Fig.

5b). The cutoff C1 and cyclone L1 aligned vertically in an equivalent barotropic manner, and precipitation fell underneath this

cutoff-cyclone couple (Fig 5b). Within the next two days, parts of the high-PV air of streamer S2 were absorbed into cutoff

C1, which was thereby substantially enlarged (Fig. 5c). At the same time, a strong anticyclone formed over northern Europe

and a next wave breaking event (S3) occurred at its downstream flank on 5 and 6 May (Fig. 5c,d). The resulting PV streamer

S3 also produced two small cutoffs (the more southerly one is labelled C2 in Fig. 5d), which tracked westward (Fig. 5d,e) and

ultimately merged with C1 on 9 May (Fig. 5f). Between 5 and 9 May, a Rex-type blocking pattern (Rex, 1950) was present

over Europe and the Mediterranean. The cutoff-complex C1 hereby acted as the positive PV anomaly on the equatorward side

of the blocking pattern, covered large parts of the Mediterranean, destabilized the air underneath and led to (primarily daytime,

e.g., Fig. 2c) precipitation from Iberia all the way to Turkey (Figs. 5c–f). Between 9 and 12 May, the cutoff-complex C1

weakened, but still sufficient precipitation fell at 12°E/43°N to prolong the wet spell there for another three days, until it finally

ended on 12 May. By 12:00 UTC 13 May, C1 had decayed entirely, although a next cutoff (C3) already approached from the

west. Interestingly, only one surface cyclone (L1) was involved in the 12-day spell $S_1(12°E, 43°N)$, and was only present

during roughly the first half of the spell. Thus, the upper-level dynamics of the cutoff-complex C1, its formation, quasi-

stationarity and replenishment due to recurrent wave breaking events, are key to this spell.

The 12-day spell $S_1(12°E, 43°N)$ is perhaps surprising in that it featured only one cyclone, despite its relatively long duration.

However, a small number of cyclones is a common characteristic of the $S_{20}(12°E, 43°N)$, as 11 of them featured two or less





**Figure 5.** As Fig. 3 but for the longest wet spell at 12°E/43°N in Tuscany, Italy [$S_1(12°E, 43°N)$]. The light red lines depict 2 PVU on 320 K. The valid time of each panel is shown in the top left.

cyclones. The cutoff-complex discussed above was tracked as an individual, long-lived system by the Portmann et al., (2021)

algorithm, but two cutoffs that merged with the cutoff complex (visible e.g., in Fig. 5d) increased the cutoff count of this spell to three. On average, 2.3 distinct cyclones and 1.9 distinct cutoffs occurred at 12°E/43°N during the $\boldsymbol{S}_{20}$. The composite 320 K PV field during the $\boldsymbol{S}_{20}(12°E, 43°N)$ is consistent with the synoptic evolution of $S_1(12°E, 43°N)$ (Fig. 7c), with an amplified trough upstream of 12°E/43°N and a wide ridge across northeastern Europe. However, at this grid point the $\boldsymbol{S}_{20}$ contain spells





with strongly differing synoptic storylines, e.g., the fifth longest spell at 12°E/43°N contrasts the longest one by occurring in
winter and by featuring five distinct cyclones and five distinct cutoffs that were steered towards 12°E/43°N by a southward
displaced jet over the eastern North Atlantic (not shown).

### 3.2.4 Long wet spell in eastern Europe – Intermittent periods of diurnal convection and recurrent cutoff formation

A fourth synoptic storyline is illustrated based on the longest wet spell at 25°E/48°N, in the Carpathian Mountains at the border
between Romania and Ukraine (Figs. 2d and 6). This wet spell lasted for an impressive 17 days, from 19 May to 5 June 1988
and featured several days with substantial precipitation without the presence of any cutoff or cyclone near 25°E/48°N. During
intermitted periods of this spell a total of three cutoffs appeared in the vicinity of 25°E/48°N, which also seemed to contribute
to the persistence of the spell. At 12:00 UTC on 20 May 1988 a breaking wave (S1 in Fig. 6) was present over western Europe,
while a surface anticyclone was located over western Russia. In-between, an area of weak SLP gradients extended across much
of the Balkans and eastern Europe, and precipitation occurred over widespread areas in this region (Fig. 6a). At 25°E/48°N the
ERA-Interim precipitation on the first two days of the spell (19 and 20 May) fell exclusively between 06 and 18 UTC (Fig.
2d), consistent with primarily convective precipitation. Over the course of the next four days, the PV feature (S1) propagated
slowly eastward (not shown) and eventually broke up into several remnants by 12:00 UTC on 24 May, including a cutoff C1
which covered 25°E/48°N at that time step (Fig. 6b). Between 21 May and 24 May, precipitation at 25°E/48°N exhibited less
of a diurnal cycle (Fig. 2d), likely due to the influence of the upper-level PV feature which continuously destabilized the
troposphere and provided quasi-geostrophic forcing for ascent. From 24 to 26 May the cutoff C1 hoovered in the vicinity of
25°E/48°N, but gradually weakened until its dissipation on 27 May (Fig. 6b–d). At 12:00 UTC on 27 May a flow situation
very much reminiscent of that on 12:00 UTC 20 May had established, with a PV streamer S2 approaching 25°E/48°N, and
widespread precipitation around 25°E/48°N without the apparent influence of a cyclone or cutoff (Fig. 6d). The streamer S2
again formed a cutoff (C2), which approached 25°E/48°N by 12:00 UTC on 29 May (Fig. 6e) and dissipated thereafter until
12:00 UTC on 30 May (not shown). It is noteworthy that during the first 13 days of the spell (19–31 May 1988) no surface
cyclone is apparent in the immediate vicinity of 25°E/48°. The large-scale flow situation changed considerably between 29
May and 1 June (Fig. 6e,f): a next PV-streamer (S3) approached 25°E/48°N, this time with a developing surface cyclone L1
to its east (Fig. 6f). The cyclone L1 deepened from 12:00 UTC on 1 July to 18:00 UTC on 2 June (Fig. 6g) and passed over
25°E/48°N, which led to the largest daily precipitation accumulations during the entire spell (Fig. 2d). By 12:00 UTC on 4
June, the streamer S3 had formed yet another cutoff (C3 in Fig. 6h), which slowly propagated eastward together with L1, and
the spell finally ended on 5 June.

The synoptic storyline of this spell is interesting in two regards: Firstly, during the first part precipitation fell due to diurnal
convection, and the influence of surface cyclones or PV cutoffs appeared to be modest at best. Secondly, during the remainder
of the spell it involved multiple cutoffs and PV streamers. Based on this case study, the intermittent occurrence of days with
daytime convection in absence of direct upper-level forcing, alternated by days with recurrent wave breaking and cutoff



formation thus emerges as a further archetypal storyline for unusually long wet spells. It should be noted that a similar synoptic storyline has recently been reported for a multi-week period of recurrent convective events in central Europe (Mohr et al., 2020).

**Figure 6.** As Fig. 3 but for the longest wet spell at 25°E/48°N in Romania [$S_1(25°E, 48°N)$]. The light red lines depict 2 PVU on 320 K. The valid time of each panel is shown in the top left.



The composite 320 K PV-field for all $S_{20}(25°E, 48°N)$ reveals a similar pattern as e.g., in Fig. 6a,c,d,e,h, with a wide ridge over Europe. The cyclone fraction, $F_{cyclone}$, is 0.21, indicating that the role of cyclones for unusually long-lasting wet spells at 25°E/48°N is much more subtle than, e.g., at 14°E/66°N (Norway case, Figs. 2b, 4, 7b), where $F_{cyclone}$ is 0.51. Cutoffs occurred at 25°E/48°N during larger fractions of the $S_{20}$ ($F_{cutoff} = 0.31$) than cyclones and in slightly larger number

($N_{cutoff} = 2.10$ vs. $N_{cyclone} = 1.90$). However, as for 12°E/43°N (Tuscany) the variability in the cyclone and cutoff characteristics across the $S_{20}$ is large, with between zero and five (one and six) distinct cyclones (cutoffs) during individual $S_{20}(25°E, 48°N)$ spells.

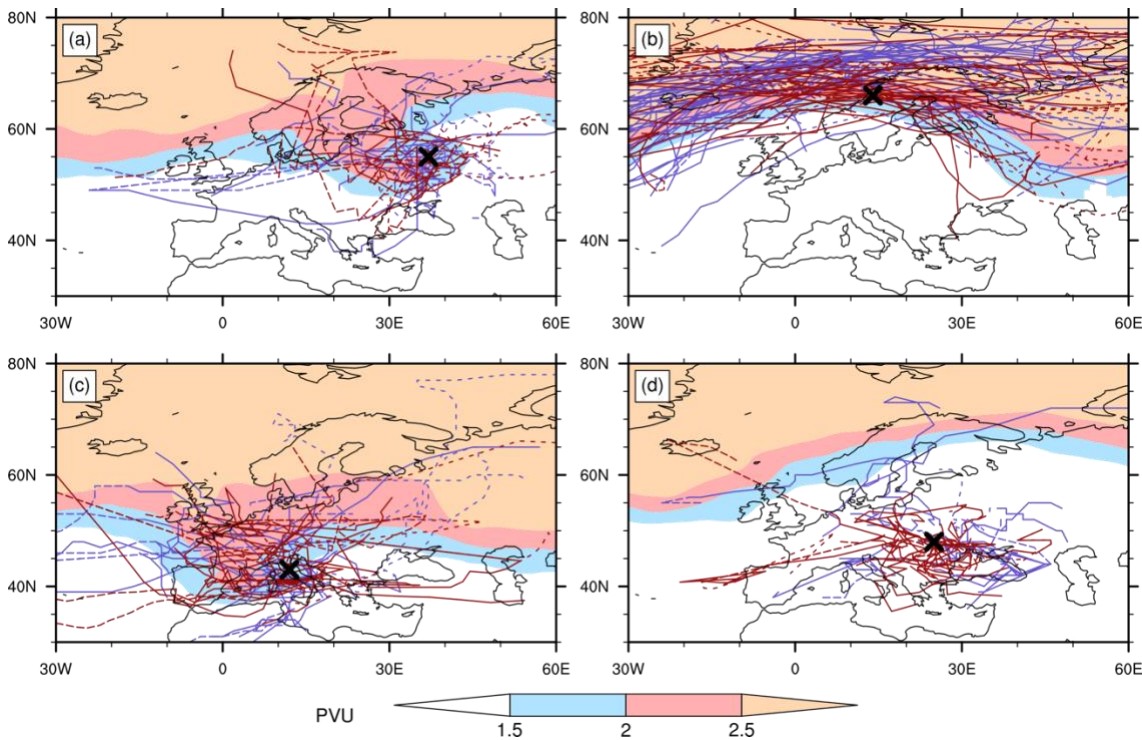

**Figure 7.** Tracks of cyclones (blue) and cutoffs (red) in the vicinity of (a) 37°E/55°N, (b) 14°E/66°N, (c) 12°E/43°N and (d) 25°E/48°N during the $S_{20}$ at the respective grid points. Cyclone and cutoff tracks are shown for cyclones/cutoffs whose mask overlapped with a circle of 400 km radius around the respective grid point for at least one 6-hourly time step during the spell. For visibility reasons, cutoff tracks are only shown, if, additionally, the center of mass of the cutoff was closer than 500 km to the respective grid point during at least one 6-hourly time step of the spell. Portions of the tracks occurring before, during, and after the respective spell are shown with long-dashed,
solid, and short-dashed lines, respectively. Composite PV fields during the respective $S_{20}$ are shown in shading on (a,c,d) 320 K, and (b) 310 K.





### 3.3 A systematic analysis of cyclone and PV cutoff characteristics during the $S_{20}$

We next examine the occurrences of cyclones and cutoffs during the $S_{20}$ across Europe more systematically to elucidate
geographical differences in their role for generating unusually long-lasting wet spells (the left columns in Figs. 8 and 9).
Furthermore, we assess whether or not the behaviour of cyclones and cutoffs during the $S_{20}$ is locally anomalous (right
columns in Figs. 8 and 9). We begin by discussing the cyclone fractions, $F_{cyclone}$, and their anomalies (Fig. 8a,b). Cyclones
occur during more than 60% of the $S_{20}$ time steps over vast parts of the North Atlantic, Scandinavia as well as the northern
and eastern Mediterranean (Fig. 8a). Moreover, $F_{cyclone}$ is generally larger over the ocean than over land, with exceptions over
large parts of Scandinavia and the UK. In the Balkans, the Caucasus and parts of the Alps, however, cyclones occur during
only roughly 10-30% of the $S_{20}$ time steps, suggesting that they do not play a major role in the longest wet spells there. The
$F_{cyclone}$ anomalies (Fig. 8b) are positive and statistically significant almost everywhere, with largest anomalies in the
Mediterranean and the subtropical North Atlantic. Insignificant $F_{cyclone}$ anomalies are found in the aforementioned areas of
particularly low $F_{cyclone}$ (which is consistent with the Balkan case study, Section 3.2.4), and, interestingly, also the west coast
of Norway.

The Norway case study (Section 3.2.2) revealed a large number of cyclones contributing to the longest spell at 14°E/66°N,
suggesting that serial clustering of extratropical cyclones (Pinto et al., 2014; Priestley et al., 2017a; Dacre and Pinto, 2020)
might be crucial for the occurrence of unusually long wet spells in this region. Figure 8e now reveals that in this region the
cyclone period, $P_{cyclone}$, is indeed shorter than anywhere else in Europe, with values between 1.5 and two days. Also,
anomalously large $N_{cyclone}$ and anomalously short $P_{cyclone}$ and $R_{cyclone}$ around 14°E/66°N support the hypothesis of serial
cyclone clustering as cause of long wet spells there. However, none of the four quantities' anomalies are statistically significant
at 14°E/66°N, a result that will be further discussed in Section 4. Clearer indication for anomalous serial cyclone clustering as
a cause of the longest wet spells is found e.g., across central Europe, in southwestern Scandinavia and the United Kingdom
(UK), where $F_{cyclone}$ is increased due to significantly more distinct cyclones (positive $N_{cyclone}$ anomalies) occurring at higher
rate (reduced $P_{cyclone}$), with $R_{cyclone}$ relatively close to climatology.

Cyclone characteristics (and their anomalies) during the $S_{20}$ that clearly contrast those in central Europe and the UK are found
in the seas around Italy. These seas feature some of the largest cyclone residence times ($R_{cyclone}$ in excess of two days, Fig.
8g) and the largest significant anomalies in $R_{cyclone}$ (more than 0.5 days, Fig. 8h) anywhere in the study region. Notably in
the seas around Italy, $N_{cyclone}$ and $P_{cyclone}$ do not differ significantly from climatology (Fig. 8d,f) despite $F_{cyclone}$ anomalies
of up to 40%, which corresponds to roughly a doubling of $F_{cyclone}$ during the $S_{20}$ compared to climatology (compare Figs. 8a
and b). These large $F_{cyclone}$ anomalies thus come about primarily due to increased $R_{cyclone}$, i.e., anomalously persistent, i.e.,



slower moving and/or longer-lived cyclones compared to climatology. A further region with similar and also significant

cyclone characteristic anomalies during the $S_{20}$ are the Baltic Seas.

**Figure 8.** Cyclone characteristics during the $S_{20}$ (left column), and their respective anomalies (right column, see Section 2.4 for the technical definition of the four quantities and their climatological values). (a,b) the cyclone fraction, $F_{cyclone}$, i.e., fraction of total $S_{20}$

time steps with a cyclone, (c,d) number of distinct cyclones per $S_{20}$ spell, $N_{cyclone}$, (e,f) the cyclone period, $P_{cyclone}$, i.e., summed $S_{20}$ duration divided by number of distinct cyclones, and (g,h) the residence time, $R_{cyclone}$, i.e., the average time each distinct cyclone affects one of the $S_{20}$ spells. Hatching in panels (b,d,f,h) depict grid points for which the anomalies are deemed statistically significant [see Section 2.4 for details].

**480**



**Figure 9.** As Fig. 8 but for cutoffs.

Next, we discuss the four quantities and their anomalies for cutoffs (Fig. 9) and contrast them with results for cyclones (Fig. 8). The cutoff fraction, $F_{cutoff}$, as well as the number of distinct cutoffs per $S_{20}$ spell, $N_{cutoff}$, are considerably smaller than $F_{cyclone}$ and $N_{cyclone}$ (compare Fig. 9a,c with Fig. 8a,c). The $F_{cutoff}$ and $N_{cutoff}$ anomalies are positive wherever they are

**485** large, but overall they are considerably smaller than those of $F_{cyclone}$ and $N_{cyclone}$, and only the $F_{cutoff}$ anomalies are significant over large coherent areas. Largest values of $F_{cutoff}$ are found in the western Mediterranean, where they significantly exceed the respective climatological values by roughly a factor of two (Fig. 9a,b). The cutoff period $P_{cutoff}$ reveals considerable geographical variations (Fig 9e), with shortest cutoff periods around 2 days in the Norwegian Sea, but as for $N_{cutoff}$, the $P_{cutoff}$ anomalies are not significant, which suggests that the rate at which cutoffs occur during the $S_{20}$ does

not differ strongly from its climatological value (Fig. 9d,f). The most striking result for cutoffs, though, is the significantly increased residence time $R_{cutoff}$ in the western Mediterranean. There, $R_{cutoff}$ values in excess of two days are observed, which is more than 0.5 days more than the climatological value. Thus, the anomalously large $F_{cutoff}$ during the $S_{20}$ in this region predominantly results from increased residence times of cutoffs, i.e., persistent cutoffs that are either slower-moving, longer-lived or both. A similar behaviour of cutoffs during the $S_{20}$ is found over the northern Black Sea as well as the most

southwestern corner of our study domain, over the subtropical North Atlantic.

Over land, comparatively long absolute $R_{cutoff}$ values (up to 1.75 days) are found in the Balkans and in north-eastern Europe (Figs. 9g), suggesting that in these regions, the cutoffs involved in the $S_{20}$ are more persistent than, e.g., those involved in the $S_{20}$ along the Norwegian coast or in western Europe (note, however, that the $R_{cutoff}$ anomalies are insignificant in all of these

regions). Nevertheless, throughout Europe's land area $F_{cutoff}$ is below 50% almost everywhere (Fig. 9a), indicating that during at least half of the total $S_{20}$ time no cutoff is present within a 400 km radius.

In summary, this section reveals geographically varying and, in some regions, locally anomalous behaviour of cyclones and cutoffs during the $S_{20}$. $F_{cyclone}$ is anomalously large during the $S_{20}$ almost everywhere, however, the causes of these positive

cyclone frequency anomalies differ in space. Increased numbers of cyclones during the $S_{20}$ explain the positive $F_{cyclone}$ values along the north-western Atlantic coast as well as in central Europe, while over the Mediterranean anomalously large residence times of cyclones are the reason for positive $F_{cyclone}$ anomalies. The Mediterranean is also the region where $F_{cutoff}$ deviates most from its climatological value, which is caused by anomalously persistent cutoffs during the $S_{20}$. Elsewhere, the characteristics of cutoffs during the $S_{20}$ vary in space, but they do not appear to be significantly different than the

climatological cutoff characteristics.

## 4    Discussion

The synoptic storylines for unusually long wet spells presented in Section 3.2 feature individual stalling cyclones and cutoffs as well as multiple recurrent such weather systems [as anticipated by Moore et al., (2021)], although some storylines are more complex and also involve daytime convection over complex topography without apparent upper-level forcing. The four case

studies were selected due to their archetypal nature, however, manual analyses of a large number of further long wet spells revealed numerous storylines that combined various features of the four archetypal storylines (e.g., multiple stationary cutoff-cyclone couples or multiple recurrent cyclones preceding a particularly stationary cutoff, etc.). Furthermore, this manual analysis revealed that distinct wet spells at a single location often do not follow the same synoptic storyline. This diversity raises the question whether or not these synoptic storylines at all stratify according to geographical regions, which motivated

our climatological analysis of cutoff and cyclone characteristics during the $S_{20}$ presented in Figs. 8 and 9.





The two columns in Figs. 8 and 9 address distinct research questions: The left column assesses how cyclone and cutoff characteristics during the $\boldsymbol{S}_{20}$ vary in space and thus addresses research question 3) in the Introduction. This information is valuable, since the general relevance of cyclones and cutoffs for precipitation is well established (e.g., Hawcroft et al., 2012;

Portmann et al., 2020) and thus, e.g., spatial variations in $F_{cyclone}$ point to a spatially varying relevance of cyclones for the $\boldsymbol{S}_{20}$. Nevertheless, the spatial variations in these cyclone and cutoff characteristics during the $\boldsymbol{S}_{20}$ are governed in part by climatological characteristics of cyclones and cutoffs. Therefore, the right column in Figs. 8 and 9 also compares the cyclone/cutoff characteristics during the $\boldsymbol{S}_{20}$ to climatological values, in order to identify anomalous weather system behaviour during the $\boldsymbol{S}_{20}$ (and thus address research question 4).


The left columns in Figs. 8 and 9 reveal that the relevance of cyclones and cutoffs for the $\boldsymbol{S}_{20}$ indeed varies greatly across space. For example, the range of $F_{cyclone}$ values with lowest values below 0.2 and largest values above 0.9 implies that in some regions, cyclones are present during almost the entire period of the $\boldsymbol{S}_{20}$, while e.g., in the Balkans, cyclones appear to be largely irrelevant to the $\boldsymbol{S}_{20}$ there. Furthermore, $F_{cutoff}$ varies from around 0.2 (e.g., over Finland) to 0.7 (over the western

Mediterranean), indicating in particular that a major fraction of the total $\boldsymbol{S}_{20}$ duration occurs without a cutoff within a 400 km radius, even in the regions with the largest $F_{cutoff}$.

The right columns in Figs. 8 and 9 show single-signed anomalies of $F$, $N$ and $P$ almost throughout the study region for both features, with positive anomalies in $F$ and $N$, and negative anomalies in $P$. Thus, both features are more prevalent (positive $F$

anomalies), occur in larger number (positive $N$ anomalies) and at a higher rate (negative $P$ anomalies) during the $\boldsymbol{S}_{20}$ than in climatology. The anomalies of $R$ take either sign for both features, however, only relatively few (and mostly positive) anomalies are significant for both types of weather systems. The lack of statistically significant anomalies in any of the four quantities may results from three causes: (a) at some grid points, there is simply no preferred synoptic storyline of the $\boldsymbol{S}_{20}$ with a clear signature in the four cyclone and cutoff characteristics, $F$, $N$, $P$ and $R$, which may be the case for example in regions

where the $\boldsymbol{S}_{20}$ occur in different seasons. (b) The sample size (20) is relatively small for a statistical hypothesis test, consequently our Monte Carlo test has only limited power to detect significant departures from climatology (e.g., Wilks, 2011). (c) In certain regions, e.g., the west coast of Norway, the climatological precipitation variability is itself characterized by long wet spells (Zolina et al., 2013). In such regions, unusually long wet spells do not need to be associated with anomalous cyclone and cutoff characteristics, but rather with close to climatological characteristics over a prolonged period.


Nevertheless, for cyclones the anomalies of $F$, $N$ and $P$ are significant in vast areas of the study region. Comparing the right columns in Figs. 8 and 9 we find overall weaker anomalies in $F$, $N$, $P$ and $R$ for cutoffs than for cyclones, which suggests a generally weaker link between anomalous cutoff characteristics and the occurrence of the $\boldsymbol{S}_{20}$ than for cyclones. In part this





could be a consequence of the cutoff definition of Portmann et al., (2021), who defined cutoffs as three dimensional objects, which can persist as vertically shallow objects on relatively high isentropes, where their influence on lower and mid-tropospheric static stability as well as ascent is limited (see Portmann et al., 2018 for an example of such a case).

In the Mediterranean, however, $F_{cutoff}$ and $R_{cutoff}$ anomalies are of similar magnitude as the $F_{cyclone}$ and $R_{cyclone}$ anomalies, and underline the importance of persistent cutoffs and cyclones for the $S_{20}$ there. This result is consistent with several previous studies, in particular Doswell et al., (1998), who documented a 7-day heavy precipitation event in Valencia, Spain, and Portmann, (2020) who showed that the Mediterranean is the region where cutoffs contribute the most to annual precipitation anywhere on the globe. Furthermore, de Vries (2021) identified the Mediterranean as a region where the odds of extreme precipitation events are significantly increased when upper-level high-PV features are present. A novel finding of this study is that cutoffs can be key dynamical precursors to even much longer precipitation episodes, in particular if such cutoffs are replenished by multiple wave breaking events (Section 3.2.3) or when multiple recurrent cutoffs are involved (Section 3.2.4). In either case, the longevity of the wet spells appears to be linked to recurrent synoptic-scale Rossby wave dynamics (the formation, amplification and breaking of upper-level troughs). Subsequent research could therefore investigate whether or not these spells have statistically significant Rossby wave precursors, possibly in the form of recurrent Rossby wave packets (Ali et al., 2021; Röthlisberger et al., 2019).

The finding that long wet spells along the west coast of Europe associated with multiple recurrent cyclones (e.g., Section 3.2.2) is not surprising, in particular given the clear evidence for serial cyclone clustering in the eastern North Atlantic (Mailier et al., 2006; Pinto et al., 2014; Priestley et al., 2017a; Dacre and Pinto, 2020). Indeed, the Norway case study (Section 3.2.2) features several aspects of the archetypal pathway to cyclone clustering outlined in Priestley et al., (2017a), with recurrent anticyclonic wave breaking over the North Atlantic on the equatorward side of the jet (e.g., Fig. 4b,c), a zonally extended and northward displaced jet (not shown) that leads to strong and persistent westerly flow impinging on Scandinavia and steers cyclones towards northern Scandinavia (Fig. 4). Furthermore, as in Priestley et al., (2017a), the cyclone clustering period is terminated when a cyclonic wave breaking event displaces the North Atlantic jet southward (Fig. 4h). Perhaps more surprising is the lack of significantly positive cyclone frequency anomalies along the northern Norwegian coast (Fig. 8b), which likely points to the relevance of a stagnant westerly moist air flow that supports orographic precipitation during the $S_{20}$ there. Furthermore, the moderate and insignificant anomalies of $N_{cyclone}$ and $P_{cyclone}$ in Norway (close to 14°E/66°N) are consistent with our explanation (c) above, i.e., serial clustering of cyclones is a climatological characteristic of cyclones in the eastern North Atlantic (Mailier et al., 2006), thus, it is certainly relevant to the generation of long wet spells there, but not an anomalous behaviour of cyclones there.

The results of and conclusions from this study are limited in a number of ways. Firstly, we use precipitation from ERA-Interim as opposed to observed precipitation (remotely sensed or in-situ measured). Consequently, the identification of the $S_{20}$ is




affected by precipitation biases in ERA-Interim (see also Section 2.1). However, using ERA-Interim precipitation for the spell identification is convenient as it is spatially and temporally complete and, furthermore, ensures that the precipitation is

physically consistent with the synoptic-scale dynamics accompanying the long wet spells studied here. Secondly, the criteria for defining "wet spells" are to a certain degree subjective and, consequently, our definition of wet spells as consecutive days with at least 5 mm daily accumulated ERA-Interim precipitation strongly affects which events ultimately end up in the $\mathbf{S}_{20}$. Nevertheless, choosing a higher precipitation threshold than previous studies (e.g., Ali et al., 2021; Zolina et al., 2010, 2013) ensures that our $\mathbf{S}_{20}$ are unusually long-lasting periods of sustained relatively intense precipitation, and thus potentially high-

impact events. Thirdly, our analyses focus primarily on only two types of synoptic systems, cutoffs and cyclones. The choice of these two systems is motivated by previous studies, who documented the particular relevance of these two types for long-lasting heavy precipitation episodes (Doswell et al., 1998; Raveh-Rubin and Wernli, 2015; Moore et al., 2021), and our case studies further underline this pivotal role of cutoffs and cyclones for long wet spells. Subsequent research should nevertheless explore to what extent other weather systems such as stagnating atmospheric rivers (e.g., Moore et al., 2021) or blocking (e.g.,

Mohr et al., 2020) foster the occurrence of unusually long wet spells in their vicinity.

## 5 Summary and conclusions

This study investigates the role of cyclones and PV cutoffs for the formation of unusually long wet spells in Europe, which are identified at each ERA-Interim grid point as the 20 longest uninterrupted periods with at least 5 mm daily accumulated ERA-Interim precipitation ($\mathbf{S}_{20}$). The $\mathbf{S}_{20}$ are longest along the Norwegian coast and northern Scotland, where the average duration

of the $\mathbf{S}_{20}$ reaches up to two weeks. The $\mathbf{S}_{20}$ are shortest e.g., in Poland and north-eastern Scandinavia, where their average duration is only 3-5 days. There is a clear seasonality associated with the occurrence of the $\mathbf{S}_{20}$: over eastern continental Europe they occur predominantly in summer, while over the North Atlantic most of the $\mathbf{S}_{20}$ occur in winter, and the majority of the $\mathbf{S}_{20}$ along European coastal seas and the Mediterranean occur in winter or fall. In central and western Europe, no season clearly dominates.


Four case studies reveal distinct synoptic storylines of selected long wet spells, that each involve cyclones and/or PV cutoffs in distinct ways. The longest wet spell in Moscow occurred in association with just one cutoff-cyclone couple that formed from a single wave breaking event, subsequently stalled over western Russia and thus produced >5 mm/day precipitation at this grid point for seven consecutive days. An even longer spell (12 days) associated with a single cutoff-complex occurred in

Tuscany, Italy. Initially, this cutoff-complex also formed from a wave breaking event but, in contrast to the Moscow case, was then replenished multiple times by multiple further wave breaking events over the North Atlantic and Europe. The Tuscany case thus illustrates recurrent wave breaking and subsequent cutoff replenishment as a further, hitherto not documented synoptic storyline for unusually long wet spells. In contrast, a substantial body of literature documents the tendency for North Atlantic extratropical cyclones to serially cluster in the North Atlantic region (Dacre and Pinto, 2020 and references therein).

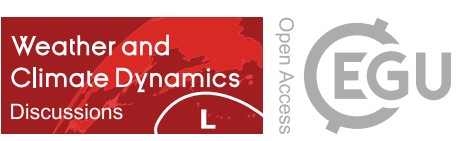

Here we document such behaviour of extratropical cyclones during a 15-day wet spell in Norway, which also involved orographic precipitation in westerly onshore flow. Finally, a 17-day wet spell in the Balkans reveals the importance of diurnal convection for this long summer-time wet spell over continental Europe, and also featured recurrent cutoffs that occur intermittently with periods of diurnal convection in absence of evident upper-level forcing.

To compare cyclone and cutoff characteristics during the $S_{20}$ in space as well as to climatological cyclone/cutoff characteristics we compute four quantities for both types of weather features and for each grid point $(x, y)$, which each consider cyclones and cutoffs that overlap with a radius of 400 km around $(x, y)$ during the $S_{20}(x, y)$: The cyclone/cutoff fraction $F$, the number of distinct cyclones/cutoffs per spell, $N$, the cyclone/cutoff period, $P$, and the cyclone/cutoff residence time $R$. This analysis reveals regionally strongly varying roles of cutoffs and cyclones for the $S_{20}$. For instance, The absolute values

of $F_{cyclone}$ vary from less than 0.2 in the Balkans (i.e., cyclones almost never present during the $S_{20}$) to more than 0.8 over the North Atlantic (i.e., the $S_{20}$ occur under quasi-continuous influence of cyclones). For both weather features, the anomalies of $F$, $N$ and $P$ are single-signed (or small) throughout Europe and imply that, during the $S_{20}$, cyclones and cutoffs are more prevalent (positive $F$ anomalies), occur in larger number (positive $N$ anomalies) and at a higher rate (negative $P$ anomalies) compared to climatology. Hereby larger anomalies of $F$, $N$ and $P$, with higher statistical significance, are found for cyclones

than for cutoffs in most regions, which suggests a tighter association between anomalous weather system characteristics and long wet spells for cyclones than for cutoffs. An exception to this rule is found in the western Mediterranean, where cutoffs have strongly and statistically significantly increased residence times during the $S_{20}$.

We conclude that the synoptic storylines accompanying unusually long wet spells are highly diverse across Europe. Cutoffs

and cyclones were involved in all cases we analyzed, either as individual and unusually persistent systems or in a recurring manner. A novel finding of this study is that recurrent Rossby wave breaking may act as an indirect precursor to unusually long wet spells, by replenishing an existing cutoff which subsequently fosters the occurrence of an unusually long wet spell. Subsequent research should therefore investigate how unusual recurrence and unusual longevity of synoptic systems comes about and how the two are intertwined. Furthermore, given the potential impacts of unusually long wet spells as well as their

potential changes in a warming climate (e.g., Pfleiderer et al., 2019), it is important to investigate whether or not climate models are able to realistically reproduce the synoptic characteristics of such events, in order to assess the reliability of projected changes in wet spell characteristics. This study documents for the first time the palette of synoptic storylines accompanying unusually long wet spells across Europe, and thus forms a basis for such climate model evaluations.

*Data availability.* ERA-Interim data can be downloaded from the ECMWF webpage (https://apps.ecmwf.int/datasets/data/interim-full-daily/levtype=sfc/).



*Author contributions.* BS and MR performed the analyses and BS produced earlier versions of the Figures 1, 3, 4 and 7–9 as part of her MSc Thesis research. MR conceived the study, produced the final figures and wrote the manuscript, RP and AJdV discussed intermediate results, and commented earlier versions of this manuscript.


*Competing interests.* The authors declare no conflict of interest.

*Acknowledgements.* All authors would like to thank Heini Wernli (ETH Zürich) for helpful discussions, feedback on an
earlier version of this manuscript and his general support for this study. MR acknowledges funding of the INTEXseas project from the European Research Council (ERC) under the European Union's Horizon 2020 research and innovation programme (grant agreement No 787652), and AJdV funding of the Swiss National Science Foundation (grant no. 177996).



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
