# Peer review of "The role of cyclones and PV cutoffs for the occurrence of unusually long wet spells in Europe"

_Weather and Climate Dynamics, 2022_

## Referee Comment (RC1)

Review of *'The role of cyclones and PV cutoffs for the occurrence of unusually long wet spells in Europe'* by Röthlisberger et al.

**Overview**

This paper presents an assessment of long wet spells across Europe and their association with PV cutoffs and extratropical cyclones. I find this an interesting study that contributes to the breadth of knowledge of extreme event drivers, although provides only small amounts of new scientific insight that has not been documented in the already published literature.

Overall, I have very few comments as this study presents four case studies and the dynamics surrounding them. My main query is regarding to the anomalies presented in Figs 8 and 9. I find the method of calculating the climatology unusual and expand more upon this below. Furthermore, I question the use of ERA-Interim reanalysis when the newer and higher resolution ERA5 has been readily available for some time now.

Once the authors address my comments I recommend this manuscript for publication as I believe it will suit the journal well. My main points to be addressed can be found below.

**Comments**

L50 – the reference needs re-formatting. The comma should not be there.

L55-65 – I would re-phrase this paragraph. The way it is introduced suggests that features such as WCB, fronts, cyclones are individual features, when this is rarely the case and they are often all part of one synoptic system. I appreciate the authors do mention this toward the end of the paragraph, however I think this could be phrased better.

L135-143 – The choice of ERA-Interim as an analysis dataset is a confusing one. Newer reanalysis products such as ERA5 have been readily available for several years now and using a more up-to-date product, with higher resolution would surely be beneficial for a study such as this. The specific dynamics and features that would be resolved would increase and also the issues with precipitation mentioned by the authors may be reduced.

Have the authors tested their selection of the wet spells to the different precipitation products? Would there be different climatologies in Figs. 1, 7, 8, 9 as a result?

L175 – How sensitive are the results to the choice of mask radius/distance from gridpoint? Why did the authors choose 400km?

L194-201 – I find the choice of how the climatologies created confusing. From my interpretation you take all the days of the year that the wet spells occur (from start to end) and create the climatology based on those days of the year? Firstly, how many days of the year are in the climatology of each grid point – surely this varies depending on the average length of the spell and how likely the spells are to overlap/be in the same season. Secondly, would it make sense to have the climatology for all wet days and then the anomalies would be for how the unusually wet days differ from just wet days? On this, the wet spells in summer are also likely averaging some significantly warm (and cyclone-less) days as well, do these skew the anomalies significantly? Is the question of the paper how do unusually long wet speels differ from wet periods, or from all other days in general? This needs to be made clearer in the introduction.

Fig. 1 – it would be good to also show the variation in the length of the extreme wet speels. How much does this variation skew the averages shown in this figure? Would the median be a better choice for some of the panels?

L294-295 (and throughout) – are the numbers quotes for $N_{cyclone}$ and $F_{cyclone}$ statistically significant? If not then this does not suggest that these wet spells feature unusual synoptic conditions.

Fig. 5 – please define the Streamers in the figure caption and the text. These are not introduced prior to this in the text and therefore should be explained.

L463-464 – I would argue that the residence times are somewhat similar for the UK and the Italian seas. I'd rephrase this paragraph to reflect the lack of differences in this field.

---

## Author Comment (AC1)

**Replies document for reviews of:**

**The role of cyclones and PV cutoffs for the occurrence of unusually long wet spells in Europe**

Matthias Röthlisberger,[1] Barbara Scherrer,[1] Andries Jan de Vries,[1,a] Raphael Portmann[1]

[1] Institute for Atmospheric and Climate Science, ETH Zürich, Zürich, Switzerland

[a] now at Institute of Earth Surface Dynamics, University of Lausanne, Lausanne, Switzerland

*Corresponding author:* Matthias Röthlisberger, matthias.roethlisberger@env.ethz.ch

**General comments to the Reviewers**

We would like to thank both reviewers for their thoughtful, encouraging and constructive comments. Below we reply to each comment and describe the changes to the manuscript resulting from them. In particular Reviewer 2 requested a substantial number of additional figures. We have therefore decided to accompany the main manuscript with a supplement, which contains the additional figures. Furthermore, two additional figures are included at the end of this document for reviewer reference. Moreover, substantial changes we made to the revised manuscript include the following. (1) We now consider additional variables in the discussion of our case studies, namely the integrated vapor transport as well as the quasi-geostrophic vertical motion at 500 hPa, forced from the atmospheric layers above 550 hPa (hereafter IVT and QGω, respectively). Moreover, we currently explore whether a quantitative measure of the tropospheric static stability can be meaningfully incorporated in this study. (2) We discovered that the plotting routine we used to produce the original Figs. 8 and 9 (NCL) only drew a hatching (to indicate statistical significance) when multiple neighbouring grid points were deemed significant. This led to the impression that many $F$, $N$, $P$ and $R$ values were not significant while in fact they should have been labelled as significant in the original Figs. 8 and 9, based on our non-parametric test. We now plot the significance information by masking out insignificant grid points in the revised Figs. 8 and 9 as well as in Figs. S6–9.

All other comments have also been addressed and were particularly useful to more clearly present our results. Reviewer comments are included below in blue font colour and our replies in black.

**Reviewer 1**

This paper presents an assessment of long wet spells across Europe and their association with PV cutoffs and extratropical cyclones. I find this an interesting study that contributes to the breadth of knowledge of extreme event drivers, although provides only small amounts of new scientific insight that has not been documented in the already published literature. Overall, I have very few comments as this study presents four case studies and the dynamics surrounding them. My main query is regarding to the anomalies presented in Figs 8 and 9. I find the method of calculating the climatology unusual and expand more upon this below. Furthermore, I question the use of ERA-Interim reanalysis when the newer and higher resolution ERA5 has been readily available for some time now. Once the authors address my comments I recommend this manuscript for publication as I believe it will suit the journal well. My main points to be addressed can be found below.

Comments

1. L50 – the reference needs re-formatting. The comma should not be there.

Ok, changed as requested, thank you for spotting this typo.

2. L55-65 – I would re-phrase this paragraph. The way it is introduced suggests that features such as WCB, fronts, cyclones are individual features, when this is rarely the case and they are often all part of one synoptic system. I appreciate the authors do mention this toward the end of the paragraph, however I think this could be phrased better.

This paragraph has been rephrased to more clearly emphasize that these features are dynamically related and often occur in association with one another.

3. L135-143 – The choice of ERA-Interim as an analysis dataset is a confusing one. Newer reanalysis products such as ERA5 have been readily available for several years now and using a more up-to-date product, with higher resolution would surely be beneficial for a study such as this. The specific dynamics and features that would be resolved would increase and also the issues with precipitation mentioned by the authors may be reduced. Have the authors tested their selection of the wet spells to the different precipitation products? Would there be different climatologies in Figs. 1, 7, 8, 9 as a result?

We see the rationale of this comment but we still believe that we have good reasons for the choice of the used data set. We very intentionally worked with precipitation data from a reanalysis data set as opposed to observational precipitation data, because the purpose of the study is to examine the role of cyclones and cutoffs for unusually long wet spells, and thus the consistency between the precipitation data and the SLP, wind, PV fields etc. is crucial. However, we agree with the reviewer that using ERA5 data in this study would, in principle, be desirable. The main reason for choosing ERA-Interim instead, though, is that the Portmann et al. (2021) PV cutoff climatology is only available for ERA-Interim and cannot easily be adapted to ERA5, due to very large computational costs arising from the sophisticated PV cutoff tracking routine which is part of the Portmann et al. (2021) algorithm. In the following three arguments we motivate our choice for using ERA-Interim data and elaborate on why using ERA-Interim as opposed to ERA5 is not expected to affect the qualitative findings of this study.

(1) The Portmann et al. (2021) PV cutoff identification algorithm involves a sophisticated three dimensional Lagrangian tracking routine, which is based on kinematic air parcel trajectories. This tracking scheme is clearly distinct and, in our opinion, superior to other tracking routines for cutoffs (e.g., Bell and Bosart 1989; Nieto et al. 2005; Pinheiro et al. 2017; Muñoz et al. 2020), for three reasons. First, it is the only one that uses the PV framework, while others are based on relative vorticity and/or geopotential height. Second, the tracking uses kinematic air parcel trajectories and quasi-conservation of PV on isentropic surfaces. These two reasons render this approach particularly consistent with fundamental principles in atmospheric dynamics. In addition, this trajectory-based approach also works in regions where cutoffs move rapidly, for example near the jet stream where consecutive features do not always overlap spatially. And third, it allows for three dimensional feature tracking and therefore circumvents any dependence on the choice of a vertical level. This is important because cutoffs often strongly evolve in their vertical structure (e.g., Portmann et al. 2018) and can therefore only be meaningfully tracked with a three-dimensional tracking scheme. However, this tracking scheme is computationally very expensive already for ERA-Interim, and applying it to ERA5 would further increase the computational costs by a factor of 24 (6 times higher temporal resolution, four times more grid points per model level), which we do not consider feasible at the current stage.

(2) We expect that synoptic to large-scale flow structures such as cyclones and upper-level PV cutoffs are well resolved already in ERA-Interim, as the major improvement from ERA-Interim to ERA5 lies in the resolution of smaller-scale processes and weather features. Note that throughout the manuscript we investigate these synoptic to large-scale flow structures and not their smaller-scale substructure. Therefore, we do not expect that discrepancies in ERA-Interim and ERA5 cyclones and cutoffs would question our conclusions on a qualitative level.

(3) We only rely on the ERA-Interim precipitation field for identifying the wet spells, whose characteristics are displayed in Fig. 1. Reproducing Fig. 1 with ERA5 data (Fig. A1, at the end of this document) reveals no drastic differences between the wet spell duration, accumulated precipitation, mean precipitation rate or seasonality compared to ERA-Interim. Especially for this reason, we do not expect a major benefit from reproducing the analysis using ERA5.

In summary, we share the reviewers view that the use of ERA5 in principle would be preferable for the current study, in particular if the PV cutoff climatology of Portmann et al. (2021) were available for ERA5. However, for the three reasons above we believe that using ERA-Interim is justified given the purpose of this study.

4. L175 – How sensitive are the results to the choice of mask radius/distance from gridpoint? Why did the authors choose 400km?

We tested radii of 200 to 600 km for both weather features and now included the respective results as Figs S6–9 in the Supplemental Material. There is little qualitative sensitivity to the exact choice of the radius $r$ for this range of values. A smaller radius than 200 km or larger radius than 600 km seems unjustified based on synoptic experience. Both cyclones and cutoffs can surely induce precipitation further away than 200 km from the identified mask, e.g., along trailing cold fronts (of cyclones) or downstream of propagating cutoffs, where the quasi-geostrophic forcing for ascent can be expected to be largest. Furthermore, if $r$ is increased beyond 600 km then during some time steps most of the study domain is "allegedly" under the influence of cyclones and cutoffs and almost all precipitation would be attributed to either of these systems. This also does not seem justified, as, e.g., in summer over complex topography the majority of precipitation falls due to diurnal convection (Rüdisühli et al. 2020).

In the revised manuscript we mention our sensitivity analysis for $r$ and the results for $r = 200$ km and $r = 600$ km.

5. L194-201 – I find the choice of how the climatologies created confusing. From my interpretation you take all the days of the year that the wet spells occur (from start to end) and create the climatology based on those days of the year? Firstly, how many days of the year are in the climatology of each grid point – surely this varies depending on the average length of the spell and how likely the spells are to overlap/be in the same season. Secondly, would it make sense to have the climatology for all wet days and then the anomalies would be for how the unusually wet days differ from just wet days? On this, the wet spells in summer are also likely averaging some significantly warm (and cyclone-less) days as well, do these skew the anomalies significantly? Is the question of the paper how do unusually long wet spells differ from wet periods, or from all other days in general? This needs to be made clearer in the introduction.

Both reviewer raised concerns with regard to the computation of climatological values for $F$, $N$, $P$ and $R$ for the two weather systems. We therefore adopted a suggestion of Reviewer 2, which was to compute the climatological values simply as the mean values over the respective Monte Carlo sample. These new climatological values do not differ substantially from the original ones, but given that both reviewers found our original approach somewhat confusing we chose to adapt this simpler definition of climatological $F$, $N$, $P$ and $R$ in order to increase the clarity of our approach. The number of days contained in this new climatological values still vary in space depending on how long the $S_{20}$ are at each grid point (as correctly noticed by the reviewer), but this seems justified given the strongly differing seasonality and duration of the $S_{20}$. In the revised manuscript (last paragraph of Section 2.5) we now more explicitly mention the variable number of days in the climatology. Moreover, the new climatological values reflect climatological values for wet and dry days and we now also specifically mention this fact in the revised manuscript (last paragraph of Section 2.5). The anomalies in the original and revised Figs. 8 and 9 therefore inform about differences in cyclone/cutoff characteristics during the $S_{20}$ and average conditions during the respective days of the year. Constructing a climatology solely of wet days would be a valid alternative, which, however, would help to address a slightly different research question, namely how cyclone/cutoff characteristics differ

during the $S_{20}$ from average wet day conditions. We believe both research questions are worthwhile, but it is the first of these two research questions that we would like to address here.

6. Fig. 1 – it would be good to also show the variation in the length of the extreme wet spells. How much does this variation skew the averages shown in this figure? Would the median be a better choice for some of the panels? L294-295 (and throughout) – are the numbers quotes for N_cyclone and F_cyclone statistically significant? If not then this does not suggest that these wet spells feature unusual synoptic conditions.

To visualize this variability amongst the $S_{20}$ panels (a-d) of the new Supplemental Figure 1 now show the duration and accumulated precipitation of the longest spell per grid point ($S_1$) and the twentieth longest wet spell ($S_{20}$). Moreover, we follow the advice of the reviewer and show in the revised Fig. 1a,b the median duration of the $S_{20}$ (Fig. 1a) and the median accumulated precipitation (Fig. 1b).

Furthermore, the reviewer is right in noticing that statistically not significant $N_{cyclone}$ or $F_{cyclone}$ do not indicate unusual cyclone characteristics. However, the purpose of quantifying the four cyclone and cutoff characteristics is not just to identify significant departures from the local climatology. Rather these quantities also inform about regional differences in the roles of cutoffs and cyclones. For the latter reason the four quantities are included in the case study discussions, irrespective of whether or not they statistically significantly differ from respective local climatological values.

7. Fig. 5 – please define the Streamers in the figure caption and the text. These are not introduced prior to this in the text and therefore should be explained.

Ok, we now introduce the term PV streamer more clearly in the revised introduction, following Appenzeller and Davies (1992).

8. L463-464 – I would argue that the residence times are somewhat similar for the UK and the Italian seas. I'd rephrase this paragraph to reflect the lack of differences in this field.

Agreed, in the revised manuscript the sentence reads: "A further region with particularly noteworthy cyclone characteristics (and their anomalies) during the $S_{20}$ are found in the seas around Italy."

**Reviewer 2**

In this study, Röthlisberger et al. examine, through illustrative case studies and systematic climatological analysis, the role of cyclones and PV cutoffs for the occurrence of unusually long wet spells in Europe, defined as the 20 longest wet spells at each grid point in the ERA Interim reanalysis during 1979–2018. Overall, I found the manuscript to be well-written, and I believe that the topic has substantial scientific merit. In addition, the results may help to inform future work on predictability and climate change impacts for these events. Increased understanding of the synoptic-scale dynamical processes and weather systems that result in very long wet spells is clearly needed. The motivation for the study, the data and methods, and the results are described in a clear and concise manner. The figures are, for the most part, straightforward to interpret and support the conclusions drawn in the text.

In my review, I came up with a number of minor comments, suggestions, and questions for the authors to consider. Once these have been satisfactorily addressed, I believe that this manuscript will be acceptable for publication in *WCD*.

Comments

1. Line 88–89: A brief discussion of the dynamical link between PV streamers and cutoffs and the process of Rossby wave breaking is needed here to provide a basis for later discussions of wave breaking and the formation of PV streamers and cutoffs throughout section 3. Accordingly, a basic definition of Rossby wave breaking in the text would also be helpful.

Ok, we added a brief discussion in the introduction of the revised manuscript.

2. Line 146: "(large-scale and convective)" It would be better to explicitly state that the precipitation amounts analyzed in this study are the sum of the large-scale and convective precipitation in the ERA-Interim data.

Ok, changed as requested.

3. Line 147: Is there a particular reason why you limited the analysis to the 20 longest wet spells? Would not the statistics be more robust if you were to include, e.g., the 50 longest wet spells instead?

The reviewer is right in noticing that a larger sample of events would increase the robustness of our statistical analyses. However, the purpose of this study is to examine *unusually long* wet spells and this speaks against substantially increasing the sample size for two reasons: (1) A long wet spell can be unusual in the sense of being a rare event (hereafter "rareness criterion"), i.e., an event for which the average waiting time until a comparable event occurs (i.e., the return period) is comparatively long. This rareness criterion is important for the motivation of this study, because an event that occurs multiple times per year is much less likely to cause significant impacts than an event with a return period of several years. With the 20 longest spells in a 40-year period, we sample events whose return period typically exceeds two years. Substantially increasing the number of spells would violate the rareness criterion for an "unusually long wet spell". (2) The analyzed spells should also be exceptional with regard to their duration in comparison to all other spells at the same grid point (hereafter "exceptionality criterion"). However, in some regions in Europe the total number of multiday wet spells (based on our 5mm/day criterion) is simply too small for substantially increasing the number of analyzed spells without violating the exceptionality criterion. To make this second aspect explicit, the new Supplementary Figure 1e shows the total number of identified spells (minimum duration of two days) and reveals that this number is highly variable in space. Along the Norwegian west coast, the spell count exceeds 900, however, e.g., around Crimea or the westernmost part of the Mediterranean fewer than 200 multi-day wet spells occurred. Over most regions, though, the total spell count exceeds 200, which means that fewer than 10% of all multi-day wet spells are considered in this study. Based on Fig. S1e and the arguments made above we believe that a sample size of 20 spells is a reasonable compromise between the statistical robustness of the results and the unusualness of the spells we analyze.

Note further that we tested the sensitivity of our results to the sample size by considering the top five and top ten longest spells at each grid point. These sensitivity tests reveals no qualitative difference to the results in Figs. 8 and 9, however, as anticipated by the reviewer, the statistical significance of the results were much reduced (not shown).

4. Line 175: How was this radius determined to be suitable for this analysis? How sensitive are the results to this radius? I suspect there are situations in which cyclones or cut-offs play an

important dynamical role in a wet spell at a given location but are located farther than 400-km from the location. Of course in this type of analysis you need to choose discrete thresholds to define events/ features and to examine relationships between them. I am not arguing that you should change this radius, but I do think some discussion regarding why it was chosen would be helpful here.

We tested radii of 200, 400 and 600 km and found little qualitative differences, even though quantitatively, the value of $r$ of course strongly affects $F$, $N$, $P$ and $R$. Analogous figures to the revised Figs. 8 and 9 but for radii of 200 km and 600 km are now included in the supplement (Figs S6–9). Importantly, none of our key findings in this analysis (spatial variability of $F$, $N$, $P$ and $R$; sign- and significance/non-significance of anomalies of $F$, $N$, $P$ and $R$) are altered qualitatively when varying the radius from 200 to 600 km. The exact choice of 400 km is based on synoptic expertise, which suggests that both cyclones and cutoffs can cause precipitation more than 200 km away from the identified mask, e.g., along trailing cold fronts for cyclones. However, considering areas beyond 600 km around the identified masks as directly affected by the cyclones/cutoffs appears inappropriate too, as with such a radius almost all precipitation might be attributed to either of the two systems. For instance, it is well known that summer precipitation over complex topography is often due to diurnal convection that is not directly due to cyclones or cutoffs (e.g., Rüdisühli et al. 2020).

We now explicitly mention the robustness of these results to variations in the radius in the Section 2.5 and, as mentioned above, include analogous figures to the revised Figs. 8 and 9 but for radii of 200 and 600 km as Figs. S6–9 in the supplement.

5. Line 194–201: I find this explanation a bit confusing. It is not clear to me how climatological values for the number of distinct cyclones per spell are computed in this manner if all days of the year in any year corresponding to the S20(x,y) are grouped. Perhaps I am misunderstanding the explanation of the methodology. It might be better to use the mean from the 1000-sample Monte Carlo distribution at each grid point as the "climatological value" as the Monte Carlo approach that you apply retains information about the consecutive days comprising each individual spell in the S20 sample.

Both reviewers raised concerns with regard to how climatological values of $F$, $N$, $P$ and $R$ were originally computed. We therefore decided to adopt this reviewer suggestion and now simply compute the climatological $F$, $N$, $P$ and $R$ for both weather systems as the mean over the respective Monte Carlo samples. The anomalies in the revised Figs. 8 and 9 and Figs S6–9 were

computed with respect to these new climatologies. Moreover, we rephrased large parts of Section 2.5 (former Section 2.4) to better explain the Monte Carlo simulations as well as the computation of anomalies of $F$, $N$, $P$ and $R$.

6. Line 216: How much variability is there in the duration of the S20 cases at each location? Are there some locations where the duration is highly variable between the S20 cases?

Yes, the variability amongst the S20 is substantial, as we are sampling wet spells that are in the upper tail of the wet spell duration distribution. To illustrate this variability Supplementary Figure 1a-d now shows the duration (a,c) and accumulated precipitation (b,d) of the spells with rank 1 and 20, respectively. Moreover, in the revised manuscript we explicitly mention this large variability when discussing Fig.1.

7. Line 217: A map of the terrain elevation over the domain in Fig. 1 could aid the reader in interpreting the results.

Supplementary Fig 1f now shows the ERA-Interim topography in the study region.

8. Line 235: Are the climatological percentiles computed for all wet days in all months of the year, or do they vary seasonally based on when the wet spell occurred?

All wet days of the year. This is now mentioned explicitly in the revised Section 3.1.

9. Line 258: An explanation is needed here regarding why these four particular cases and locations were selected.

In the last sentence of Section 3.1. we explicitly state that we selected these cases subjectively due to their archetypal nature, out of a much larger set of cases we analyzed.

10. Line 260: I really appreciate the concise yet information-dense synoptic analysis and discussion for the four selected wet spells. A main criticism I have for this study is that the synoptic analysis does not include quantitative diagnostics of processes and ingredients by which the cutoffs and cyclones support the persistent precipitation. While these processes are at times inferred or surmised in the text, no diagnostics for moisture transport, forcing for ascent, convective instability are provided. Inclusion of additional fields and diagnostics for each case would, of course, result in an increase in the number of figures and in the complexity

of the manuscript, so perhaps it is outside the scope of this study. Could additional analyses and diagnostics be provided as online supplemental materials instead?

The reviewer correctly notices that excessive analyses of additional variables would push the length of the paper beyond reasonable limits. Moreover, our synoptic discussion of the case studies is based on well-known effects of cyclones and cutoffs on static stability and precipitation formation, which are also not contested by either of the reviewers. Therefore, we think it would be excessive to add another four figures to the main manuscript (one per case study) to quantify what is qualitatively apparent already in the current Figs. 3–6. Nevertheless we have compiled ERA-Interim climatologies of IVT and QGω at 500 hPa, forced from the atmospheric layers above 550 hPa as in (Graf et al. 2017) and now show these additional (quantitative) diagnostics for all four case studies at the same time steps as in Figs 3–6 as Figs. S2–5. Moreover, we have rephrased parts of Section 3.2 to also discuss these new figures. As expected, the new figures generally support quantitatively what was qualitatively evident already from the original Figs. 3–6. Furthermore we currently explore whether we can meaningfully incorporate a quantitative measure of tropospheric static stability in Figs S2–5.

11. Line 260: Perhaps this is outside the scope of your study, but I wonder if it is possible to include analysis and/or discussion of the large-scale/planetary-scale conditions that contributed to the occurrence of the four selected wet spells. Were persistent weather regimes in place over the Atlantic/Europe region that favored the synoptic-scale dynamical processes operating in each archetypal synoptic story line?

Interesting comment. We specifically and very much intentionally narrowed the scope of this study to cyclones and PV cutoffs, but of course the reviewer is right in mentioning that large-scale/planetary-scale conditions would be interesting too. However, in particular with the rather numerous other additional analyses that were requested by the reviewers, we feel that including further analyses of weather regimes and/or large-scale modes of variability would go beyond the scope of this study.

12. Line 280: It could be useful to include more fields in the composite analyses for the four locations. Possible additional fields include sea level pressure anomalies, PV anomalies, and frequency anomalies of cyclones and cutoffs. Such additional fields could provide a more detailed, complete picture of the synoptic-scale conditions conducive to the S20 cases at each location.

For the revised Fig. 7 we removed the cyclone/cutoff tracks and show instead composites of SLP anomalies, IVT anomalies and anomalies of QGω. We feel that these additional variables further clarify the composite structure of the $S_{20}$ and therefore significantly add to the value of Fig. 7. Many thanks for this comment!

13. Line 288: The information density in Figs. 3–6 is high. Overall, I think this is fine; I am able to read and interpret the figures fairly well. I do, however, recommend drawing the geographical boundaries and the SLP in different colors. This could help the reader distinguish the SLP field, especially when contours for several fields are superimposed.

Ok, we changed the color of the geographical boundaries to a lighter gray.

14. Line 297: You clearly and convincingly describe how recurrence and/or persistence of weather systems is key to the long durations of the four wet spells analyzed. I propose that Hovmoller diagrams of, say, upper-level PV anomalies or upper-level meridional wind anomalies overlaid by the cyclone and cutoff masks averaged in some latitude band would help to more clearly illustrate the recurrence and persistence during the spells. These diagrams would nicely complement the plan-view analyses in Figs. 3–6.

Interesting comment, thank you. We produced the respective Hovmöller diagrams of 250 hPa meridional wind, with feature tracks overlayed (Fig. A2, at the end of this document). For the Norway case study the Hovmöller diagram together with the cyclone tracks is indeed underlining the discussion of this case in the manuscript. There is clearly recurrent ridge formation over the North Atlantic, which is associated with the recurrent passage of (fast moving) cyclones. For the other cases, though, the meridional wind signal induced by the cutoffs/PV streamers does not come out very clearly, presumably due to the complex shape of these features (see Figs. 3, 5 and 6). Also, the cyclone/cutoff tracks do reveal to some extent the stationarity of the respective features, but due to the intricacies of feature tracking (e.g., merger and splitting events) they are difficult to interpret. We therefore think that Fig. A2 creates more confusion than clarity and refrain from including it in the main manuscript.

15. Line 343: Physically this makes sense because PV cut-offs often form in association with Rossby wave breaking that results in meridional elongation of PV streamers equatorward of the midlatitude jet/waveguide. This location is too far north to be frequently impacted by cutoffs.

Yes, we agree mostly, although not all cutoffs in the Portmann et al. (2021) climatology necessarily need to form from anticyclonic wave breaking. In fact, Portmann et al. (2021) found that, aside from the aforementioned classical storyline of cutoff formation, they also frequently form from cyclonic Rossby wave breaking associated with extratropical cyclones in the storm track regions (i.e. poleward of the jet). According to Portmann et al. (2021), the PV cutoff frequency in DJF over Norway is between 7-9%, which is not substantially less than the 9-11% in the Mediterranean region. We therefore chose to keep our original, somewhat more general wording.

16. Line 367: What processes were conducive to recurrent wave breaking/cutoff formation in this case? It seems that the recurrence was associated with temporal clustering of cyclone developments and associated ridge building upstream along the waveguide over the North Atlantic. Was this flow evolution related to an anomalous configuration of the North Atlantic waveguide? It may be worthwhile to briefly speak to the upstream processes that result in recurrent wave breaking.

Interesting and certainly valid comment. However, we do not see a very straight forward way to determine what exactly was conducive to the recurrent wave breaking. To thoroughly address this question, simulations and/or statistical analyses of a large-enough sample of similar episodes would be required. We feel that such analyses would clearly go beyond the scope of this study. Moreover, we would like to refrain from simply hypothesizing about these causes, as without the aforementioned statistical/model-based analyses, hypothesizing is really all we could do.

17. Line 430: I understand the justification for plotting all of the PV cutoff and cyclone tracks in this figure. However, I find it very difficult to make sense of the messy bundles of tracks in the maps, with the exception being Fig. 7b in which the tracks are mostly zonal. Is there a way to more clearly illustrate the track information? Alternatively, could the tracks be removed from these figures without compromising the discussion?

For the revised Fig. 7 we removed the cyclone/cutoff tracks and now show instead composites of SLP anomalies, IVT anomalies and anomalies of QGω.

18. Line 461: Can you briefly explain why the Pcyclone quantity is anomalously low nearly everywhere on the map?

409 This simply means that cyclones occur at a higher rate during the S20 compared to
410 climatology, because the rate at which distinct cyclones occur is the inverse of the cyclone
411 period $P$.

412 **Technical corrections:**

413 Line 9: Define "PV" acronym here?

414 Ok

415 Line 67: "For unusually long-lasting wet spells, it is much less clear how and in association
416 with which weather systems they form." This sentence is a bit clunky. Here is a possible
417 alternative: "The mechanisms and weather systems contributing to the occurrence of
418 unusually long-lasting wet spells are less clear."

419 Ok

420 Line 67–68: Change "Only few" to "few"

421 Ok

422 Line 109: Remove "responsible"

423 Ok

424 Line 163: Insert "the method of" before "Portmann"

425 Ok

426 Line 223: Insert "daily" before "precipitation rate"

427 Ok

428 Line 379: Start a new sentence with "The fifth"

429 Ok

430 Line 446: Insert "tend to" after "do not"

431 We prefer our original formulation.

432    Line 522: Replace the colon with a period.

433    Ok

434    Line 525: Check whether "e.g.," is needed here.

435    We think it is ok as it is.

436    Line 528: Replace "behaviour" to "characteristics" ?

437    Ok

**References**

Appenzeller, C., and H. C. Davies, 1992: Structure of stratospheric intrusions into the troposphere. *Nature*, **358**, 570–572, doi:10.1038/358570a.

Bell, G. D., and L. F. Bosart, 1989: A 15-year climatology of Northern Hemisphere 500 mb closed cyclone and anticyclone centers. *Mon. Weather Rev.*, **117**, 2142–2163, doi:https://doi.org/10.1175/1520-0493(1989)117<2142:AYCONH>2.0.CO;2.

Graf, M. A., H. Wernli, and M. Sprenger, 2017: objective classification of extratropical cyclogenesis. *Q. J. R. Meteorol. Soc.*, **143**, 1047–1061, doi:10.1002/qj.2989.

Muñoz, C., D. Schultz, and G. Vaughan, 2020: A midlatitude climatology and interannual variability of 200- and 500-hPa cut-off lows. *J. Clim.*, **33**, 2201–2222, doi:10.1175/JCLI-D-19-0497.1.

Nieto, R., and Coauthors, 2005: Climatological features of cutoff low systems in the Northern Hemisphere. *J. Clim.*, **18**, 3085–3103, doi:10.1175/JCLI3386.1.

Pinheiro, H. R., K. I. Hodges, M. A. Gan, and N. J. Ferreira, 2017: A new perspective of the climatological features of upper-level cut-off lows in the Southern Hemisphere. *Clim. Dyn.*, **48**, 541–559, doi:10.1007/S00382-016-3093-8/TABLES/1.

Portmann, R., B. Crezee, J. Quinting, and H. Wernli, 2018: The complex life-cycles of two long-lived potential vorticity cutoffs over Europe. *Q. J. R. Meteorol. Soc.*, **144**, 701–719, doi:10.1002/qj.3239.

Portmann, R., M. Sprenger, and H. Wernli, 2021: The three-dimensional life cycles of potential vorticity cutoffs: A global and selected regional climatologies in ERA-Interim (1979–2018). *Weather Clim. Dyn.*, **2**, 507–534, doi:10.5194/WCD-2-507-2021.

Rüdisühli, S., M. Sprenger, D. Leutwyler, C. Schär, and H. Wernli, 2020: Attribution of precipitation to cyclones and fronts over Europe in a kilometer-scale regional climate simulation. *Weather Clim. Dyn.*, **1**, 675–699, doi:https://doi.org/10.5194/wcd-1-675-2020.

Ventura, V., C. J. Paciorek, and J. S. Risbey, 2004: Controlling the proportion of falsely rejected hypotheses when conducting multiple tests with climatological data. *J. Clim.*, **17**, 4343–4356, doi:10.1175/3199.1.

Wilks, D. S., 2016: "The stippling shows statistically significant grid points": How research results are routinely overstated and overinterpreted, and what to do about it. *Bull. Am. Meteorol. Soc.*, **97**, 2263–2273, doi:10.1175/BAMS-D-15-00267.1.

**Additional figures for reviewer reference**

[Figure]

Fig A1: As (the revised) Fig. 1 but for ERA5. Note that contrary to the original Fig. 1 panels (a) and (b) show the median duration and median accumulated precipitation of the ERA5 $S_{20}$.

[Figure]

Fig A2: Hovmöller plots of 250 hPa meridional wind ($v$) for the four case studies (a) $S_1(37°E, 55°N)$, (b) $S_1(14°E, 66°N)$, (c) $S_1(12°E, 43°N)$ and (d) $S_1(25°E, 48°N)$. The meridional wind $v$ has been averaged in a latitude band of $\pm 15°$ latitude around the latitude of the respective spell, i.e., in (a) 40°N–70°N, in (b) 51°N–81°N, in (c) 28°N–58°N and in (d) 33°N–63°N. Horizontal lines depict the start and end dates of the respective spell, while the vertical line in each panel denote the longitude of the respective spell. Purple and yellow lines indicate the longitude–time tracks of all cyclones and cutoffs whose masks overlapped with a circle of radius 400 km around the location of the respective wet spell. In panel (b) no cutoff tracks are shown due to the insignificance of cutoffs for the synoptic storyline of this spell.

---

## Author Response (AR2)

**Replies document for second round of reviews of:**

**The role of cyclones and PV cutoffs for the occurrence of unusually long wet spells in Europe**

Matthias Röthlisberger,[1] Barbara Scherrer,[1] Andries Jan de Vries,[1,a] Raphael Portmann[1]

[1] Institute for Atmospheric and Climate Science, ETH Zürich, Zürich, Switzerland

[a] now at Institute of Earth Surface Dynamics, University of Lausanne, Lausanne, Switzerland

*Corresponding author:* Matthias Röthlisberger, matthias.roethlisberger@env.ethz.ch

**Reviewer Ben Moore**

Technical corrections:

Line 369: Consider whether subjective adverbs such as "remarkably" are needed.

Done.

Line 403: Replace "hoovered" with "remained"

Done.

Line 458: Replace "are" with "is"

Done.